# Model predictive path integral for decentralized multi-agent collision avoidance

Stepan Dergachev[1,2] and Konstantin Yakovlev[1,2]

[1] HSE University, Moscow, Russia
[2] Federal Research Center "Computer Science and Control" of Russian Academy of Sciences, Moscow, Russia

## ABSTRACT

Collision avoidance is a crucial component of any decentralized multi-agent navigation system. Currently, most of the existing multi-agent collision-avoidance methods either do not take into account the kinematic constraints of the agents (*i.e.*, they assume that an agent might change the direction of movement instantaneously) or are tailored to specific kinematic motion models (*e.g.*, car-like robots). In this work, we suggest a novel generalized approach to decentralized multi-agent collision-avoidance that can be applied to agents with arbitrary affine kinematic motion models, including but not limited to differential-drive robots, car-like robots, quadrotors, *etc*. The suggested approach is based on the seminal sampling-based model predictive control algorithm, *i.e.*, *MPPI*, that originally solves a single-agent problem. We enhance it by introducing safe distributions for the multi-agent setting that are derived from the *Optimal Reciprocal Collision Avoidance (ORCA)* linear constraints, an established approach from the multi-agent navigation domain. We rigorously show that such distributions can be found by solving a specific convex optimization problem. We also provide a theoretical justification that the resultant algorithm guarantees safety, *i.e.*, that at each time step the control suggested by our algorithm does not lead to a collision. We empirically evaluate the proposed method in simulation experiments that involve comparison with the state of the art in different setups. We find that in many cases, the suggested approach outperforms competitors and allows solving problem instances that the other methods cannot successfully solve.

## INTRODUCTION

Multi-agent navigation is a complex and challenging problem that arises in various domains such as mobile robotics, video game development, crowd simulation, *etc*. In many cases, when communication between agents is limited or not possible, the problem has to be solved in a decentralized fashion. In this case, collision avoidance is achieved through the iterative and independent finding of individual control actions by each agent, based only on local observations. Moreover, in practice, especially in robotics, agents cannot

Corresponding author
Stepan Dergachev, dergachev@isa.ru

change their movement direction (heading) instantaneously and arbitrarily. For example, consider a moving car-like robot that needs to perform a turning maneuver to change its heading. These types of constraints are known as kinematic constraints.

However, existing collision-avoidance methods either disregard such constraints (*Van Den Berg et al., 2011a*; *Zhou et al., 2017*), require extensive pre-processing to handle them (*e.g.*, creating look-up tables *Alonso-Mora et al., 2013*; *Claes et al., 2012*; *Hennes et al., 2012*), or focus only on specific kinematic motion models (*Snape et al., 2011*; *Van Den Berg et al., 2011b*; *Snape et al., 2010, 2014*).

Meanwhile, in the single-agent realm, there exists a powerful technique, *i.e.*, *Model Predictive Path Integral* (*MPPI*), that addresses the problem of computing optimal control *via* the combination of *Model Predictive Control* (*MPC*) and sampling-based optimization. The distinctive feature of *MPPI*-based methods is that they can naturally handle arbitrary nonlinear dynamics and a variety of cost functions. In the context of collision avoidance, *MPPI*-based techniques can be employed for both single-agent (*Williams et al., 2017*; *Buyval et al., 2019*) and multi-agent (*Streichenberg et al., 2023*; *Song et al., 2023*) setups by adjusting the cost function or introducing additional constraints. However, these methods do not provide safety guarantees; they can output controls that may lead to a collision.

In this work, we present a collision avoidance *MPPI*-based algorithm designed for a broad class of affine nonlinear systems that focuses on producing safe controls. Safety is achieved through a specific shift of the parameters of the distribution from which the controls are sampled. As a result, the probability that the sampled controls are within a safe subset exceeds a predefined threshold. Figure 1 showcases an illustrative example of a collision avoidance problem and corresponding solutions that involve sampling in the velocity domain and utilize either a predefined distribution (Fig. 1B) or a distribution with safe parameters derived from linear constraints (Fig. 1C). Furthermore, we demonstrate that computing the safe distribution parameters can be done by solving a *Second-Order Cone Programming* (*SOCP*) problem. By employing these new parameters, sampling efficiency is improved, and the resulting solution is guaranteed to maintain a desirable level of safety. Through extensive experimental evaluation, we validate the effectiveness of our method in successfully achieving goal positions with collision avoidance across various scenarios, including differential-drive robot dynamics and car-like dynamics. Importantly, our proposed approach outperforms existing state-of-the-art collision avoidance methods such as *ORCA for differential-drive robots* (*Snape et al., 2010*), *B-UAVC* (*Zhu, Brito & Alonso-Mora, 2022*), and learning-based methods (*Blumenkamp et al., 2022*), particularly in scenarios involving dense agent configurations, where our approach produces more efficient solutions.

The remainder of this article is organized as follows: First, we provide a comprehensive overview of the relevant literature. Subsequently, we formulate the problem statement for decentralized multi-agent collision avoidance. Furthermore, we present a background on sampling-based control and optimal reciprocal collision avoidance. The next section provides a detailed description of the proposed approach, along with a discussion of the theoretical properties. Following this, we detail the experimental setup and corresponding results, before drawing conclusions.

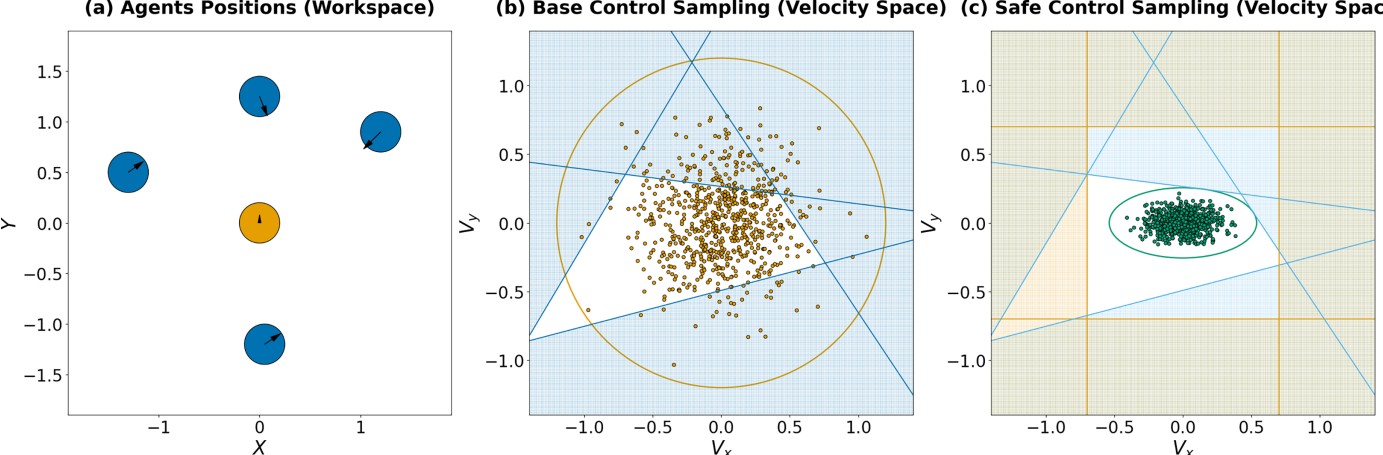

**Figure 1** (A) A visualization of an instance of the decentralized multi-agent collision avoidance problem studied in this work. Five disk-shaped agents are simultaneously moving in the environment toward their goal locations. (B) Brown dots denote controls (*i.e.*, the velocity components) sampled by the brown agent from a predefined distribution. Numerous samples lie outside a safe zone (depicted in blue) that is defined *via* the collision avoidance linear constraints. (C) Green dots correspond to controls sampled from the distribution constructed by our method. Indeed, all of them fall within the safe zone, which is additionally shrank due to the control limits (orange lines).

## RELATED WORKS

There is a wide range of methods for solving the multi-agent navigation problem. Moreover, the formulation of the problem can vary significantly in different cases. The classical formulation involves each agent aiming to reach a specific location in space, cooperating with other agents and trying to avoid collisions with them or static/dynamic obstacles (*Stern et al., 2019*). However, there are more general statements, for example, where each agent has a list of goals available, but the goal of a particular agent is not predefined, and the task is to move every agent to some goal from the list that is not occupied by other agents (this problem formulation is often referred to as anonymous or unlabeled multi-agent navigation/pathfinding) (*Stern et al., 2019*; *Solovey & Halperin, 2016*; *Turpin et al., 2014*). Another formulation, referred to as *online* or *lifelong multi-agent navigation/pathfinding*, involves agents receiving a new goal and continuing to move after the old goal is achieved (*Stern et al., 2019*; *Ma et al., 2017*).

### Centralized multi-agent navigation

Generally, navigation approaches can be divided into centralized and decentralized ones. In the centralized case (commonly referred to as *Multi-Agent Pathfinding* or *MAPF*), it is assumed that there is a central controller that creates a global plan for all agents and has reliable connections with all agents at every moment in time (*Stern et al., 2019*). Such methods usually rely on a discretized representation of the workspace (*e.g.*, a grid graph or a roadmap), although methods that consider continuous scenarios also exist (*Walker, Sturtevant & Felner, 2018*; *Andreychuk et al., 2022*). Additionally, *MAPF* algorithms often provide some theoretical guarantees, with some aimed at

obtaining optimal solutions (*Sharon et al., 2015*) or bounded-suboptimal solutions (*Barer et al., 2014*). However, these algorithms do not scale well to large numbers of agents and, in practice, do not allow for finding solutions in a reasonable amount of time.

On the other hand, if it is important to find a solution quickly, there are rule-based solvers such as *De Wilde, Ter Mors & Witteveen (2014)*. However, the solutions obtained in this case are usually far from optimal in terms of cost. A possible compromise between the solution cost and performance may be provided by prioritized planning (*Čáp et al., 2015*), which often finds close-to-optimal solutions and is also fast and scalable. Nonetheless, prioritized planning is generally incomplete.

## Decentralized multi-agent navigation

In a decentralized case, no central controller is assumed, and every agent makes decisions about its actions independently. Different works make various assumptions about the availability and quality of communication between agents. On one hand, methods can rely only on observable information, for example, knowing only the position of other agents (*Zhou et al., 2017*) or the position and velocity (*Van Den Berg et al., 2011a*). Moreover, the visibility range can also be limited (*Wang et al., 2022*). On the other hand, there are also methods that use internal information on other agents, such as planned trajectories or chosen control actions (*Zhu & Alonso-Mora, 2019*).

It is important to note that one of the basic navigation schemes used in the decentralized case is that each agent independently creates its global trajectory, taking into account only the static environment, and then moves along it, avoiding collisions with other agents. The global trajectory can also be modified, taking into account other agents, to reduce the risk of collision or avoid deadlocks (*Şenbaşlar et al., 2023*). Global trajectories can be obtained using a discrete environment representation (*Yap, 2002*) and one of the graph pathfinding algorithms, such as *Dijkstra's algorithm* (*Dijkstra, 1959*) or one of the *A\**-based heuristic search algorithms (*Hart, Nilsson & Raphael, 1968*; *Daniel et al., 2010*). The problem of single-agent path planning is out of scope for this work, but this area is well studied, and more relevant approaches can be found in *Algfoor, Sunar & Kolivand (2015)* and *Patle et al. (2019)*.

Collision avoidance methods solve the problem of following a global path while ensuring movement safety and generating only the next safe step (although they may consider several steps in advance). The collision avoidance procedure runs at a higher frequency than the global scheduler and receives the most up-to-date information (*Şenbaşlar et al., 2023*). In order to preserve the safety guarantee, such methods are required to obtain solutions taking into account as many factors as possible, such as localization or sensor data uncertainty, the agent's shape, or kinematic constraints. Decentralized collision avoidance methods will be discussed in more detail below.

## Multi-agent collision avoidance

This section will address the methods for decentralized multi-agent collision avoidance. In our work, we will consider approaches that rely on the assumption that all agents use the

same policy, act cooperatively (*i.e.*, they share responsibility for avoiding collisions), but at the same time have limited communication abilities. Existing methods can be divided into classical or model-based methods and learning-based methods. Below, we will look at the most notable model-based methods, as well as consider a learning-based approach.

### Velocity obstacles

The most well-known collision avoidance methods are based on the concept of velocity obstacles (*Fiorini & Shiller, 1998*). The main idea of such methods is to construct a safe subset in the velocity space and, at each step, find a safe velocity close to some preferred one inside such a subset. To find such subsets, velocity-based algorithms need information about the positions, velocities, and sizes of neighboring agents. The first velocity-based method that considered the problem of decentralized collision avoidance was the *Reciprocal Velocity Obstacles (RVO)* algorithm (*Van den Berg, Lin & Manocha, 2008*). Due to the reciprocity of collision avoidance, *RVO* allowed a reduction in the number of oscillations. However, this approach guarantees collision avoidance only under specific conditions, and oscillations still occur in various cases.

To eliminate these shortcomings, the *Optimal Reciprocal Collision Avoidance (ORCA)* (*Van Den Berg et al., 2011a*) and *Hybrid Reciprocal Velocity Obstacles (HRVO)* (*Snape et al., 2011*) methods were developed. The authors of *Snape et al. (2011)* mainly focused on eliminating the "reciprocal dances" effect, where agents cannot reach an agreement on which side to pass each other. On the other hand, the main contribution of the *ORCA* algorithm is that it provides a sufficient condition for each agent to be collision-free (*Van Den Berg et al., 2011a*). Moreover, the *ORCA* method allows for generating smoother trajectories than *RVO* and *HRVO* (*Douthwaite, Zhao & Mihaylova, 2019*). However, these methods find solutions in velocity space rather than action space and cannot take into account kinematic and dynamic constraints in general. In addition, the *ORCA* algorithm does not consider uncertainty in the data.

Thus, an important area of research is the adaptation of velocity-based methods to various kinematic constraints. The authors of *Van Den Berg et al. (2011b)* suggest the *Acceleration Velocity Obstacle (AVO)* approach as a generalization of the basic Velocity Obstacle method for various dynamics, but it does not exploit the advantages of reciprocity introduced in *RVO*, *HRVO*, and *ORCA* methods. *Snape et al. (2010, 2014)* describe the adaptation of the *ORCA* algorithm to the differential-drive robot dynamics by enlarging the radius of the agent, and also proposes a second adaptation approach based on the ideas of *AVO*.

The most general approach to adapting velocity-based methods to non-holonomic constraints (*Non-Holonomic ORCA* or *NH-ORCA*) is described in *Alonso-Mora et al. (2013)*. In *Alonso-Mora et al. (2013)*, the authors considered only the application of the *NH-ORCA* on differential-drive robots, but in *Alonso-Mora, Beardsley & Siegwart (2018)*, the dynamical systems of several other types were also taken into consideration. On the one hand, *NH-ORCA* presents a general scheme that allows the consideration of constraints of various types. However, it requires the preparation of special lookup tables

for each specific motion model to find the maximum tracking error of the holonomic trajectory by the non-holonomic agent. In addition, such methods find solutions in velocity space, which requires the preparation of a special controller that allows for the execution of a safe velocity with specified tracking error (*Alonso-Mora, Beardsley & Siegwart, 2018*).

Methods to take into account uncertainty in information on state and environment have also been developed, *e.g.*, *Probabilistic RVO (PRVO)* (*Gopalakrishnan et al., 2017*), *Collision Avoidance with Learning Uncertainty (CALU)* (*Hennes et al., 2012*), and *Combined Collision Avoidance with Learning Uncertainty (COCALU)* (*Claes et al., 2012*). Both *CALU* and *COCALU* also leverage the ideas of *NH-ORCA* to consider the differential-drive kinematic constraints.

### Buffered Voronoi cells

An alternative to velocity-based methods is to use approaches based on *buffered Voronoi cells (BVC)* (*Zhou et al., 2017*). The main idea of such methods is to construct the Voronoi cells around each agent or obstacle and, at each step, find an action that would keep the agent inside the cell but move it towards the goal. An important advantage of such methods is that there is no need for information on other agents' velocities. However, such methods are subject to the issue of deadlocks and oscillations. The authors of the original *BVC* method (*Zhou et al., 2017*) propose using the right-hand rule, where each agent always chooses to detour from its right side when encountering other agents. An alternative approach is to use buffer zones of varying sizes depending on the priority of the agent (*Pierson et al., 2020*), or to employ other deadlock resolution rules (*Arul & Manocha, 2021*).

Similar to velocity-based methods, accounting for uncertainty and kinematic constraints is an essential area of research, *e.g.*, *Wang & Schwager (2019)* and *Zhu, Brito & Alonso-Mora (2022)*. Let us consider the *Buffered Uncertainty-Aware Voronoi Cells (B-UAVC)* method introduced by *Zhu, Brito & Alonso-Mora (2022)* in more detail. It modifies the *BVC* method in a way that bases every action selection on the estimated position and uncertainty covariance of agents, their neighbors, and obstacles. The article also introduces approaches to adapting *B-UAVC* to several specific dynamic models, ultimately suggesting the use of *Model Predictive Control* with *B-UAVC* constraints, necessitating the development of a suitable *MPC* controller for the specific dynamics.

### Safety barrier certificates

Another approach to solving the collision avoidance problem is the use of *Safety Barrier Certificates (SBC)* (*Wang, Ames & Egerstedt, 2017*; *Luo, Sun & Kapoor, 2020*). The core idea of the method is to create linear constraints in the agent's control space and find a safe control input that is close to an optimal one. In the basic version, the algorithm requires agents to know the internal states of other agents (*e.g.*, current acceleration and control constraints). However, considering certain dynamic models and assuming some prior knowledge of agents about each other, this algorithm can be considered decentralized.

Nonetheless, it should be noted that finding safe control necessitates the development of a controller that provides optimal control based on user-defined criteria.

### Reinforcement learning approaches

A separate area of research in collision avoidance is the development of reinforcement learning-based methods (*e.g.*, *Long, Liu & Pan, 2017*; *Long et al., 2018*; *Chen et al., 2019*; *Fan et al., 2020*; *Everett, Chen & How, 2021*; *Blumenkamp et al., 2022*). The basic scheme of most methods involves approximating the function (using neural networks) that estimates actions regarding safety and progress toward the goal (the value function). At each step, the algorithm selects the most valuable action and then executes it. One significant advantage of such methods is that they are not demanding of input data and can operate using only sensor data (*e.g.*, LIDAR points), instead of the positions and velocities of other agents. However, it is essential to note that these methods often lack theoretical guarantees, and their generalizability to various scenarios and dynamic systems requires verification.

## PROBLEM STATEMENT

Consider a set of agents, $\mathcal{N} = \{1, 2, \ldots, N\}$ that operate in a shared workspace $\mathbb{R}^2$. Let $G = \{\mathbf{g^1}, \mathbf{g^2}, \ldots, \mathbf{g^N}\}$ denote agents goal locations in $\mathbb{R}^2$. Let $\mathcal{T} = 0, \Delta t, 2\Delta t, \ldots$ be the discrete time ($\Delta t = const$) and for the sake of simplicity assume that $\Delta t = 1$, thus the timeline is $\mathcal{T} = 0, 1, 2, \ldots$. The transition model of all agents is defined in non-linear affine discrete-time form:

$$\mathbf{x_{t+1}} = F(\mathbf{x_t}) + G(\mathbf{x_t})\mathbf{u_t} \tag{1}$$

where $\mathbf{x_t} \in \mathbb{R}^n$ is the agent's state at time moment $t$, $\mathbf{u_t} = (u_{1,t}, u_{2,t}, \ldots, u_{m,t}) \in \mathbb{R}^m$ is the control input at time moment $t$, $F : \mathbb{R}^n \to \mathbb{R}^n$ and $G : \mathbb{R}^{n \times m} \to \mathbb{R}^n$ are the functions that define the motion of the agent. Every component of the control vector may be bounded: $u_{k,min} \leq u_{k,t} \leq u_{k,max}$.

The state of the agent necessarily includes its position in the workspace and may include additional components (*e.g.*, orientation): $\mathbf{x_t} = (p_x, p_y, \ldots)$. Since in this work we consider a motion model with discrete time, we assume that during transition between positions $(p_{x,t}, p_{y,t})$ and $(p_{x,t+\Delta t}, p_{y,t+\Delta t})$ at adjacent moments in time $t, t + \Delta t$, the agent moves uniformly and straightforwardly.

**Example** Let us consider a differential-drive robot. In this case, the robot's state is described as $\mathbf{x_t} = (p_{x,t}, p_{y,t}, \theta_t)$, where $\theta$ is the agent's orientation (heading angle). The control input is $\mathbf{u} = (v, w)^T$, where $v$ is the linear velocity ($v_{min} \leq v \leq v_{max}$) and $w$ is the angular velocity ($w_{min} \leq w \leq w_{max}$). The functions that describe the robot's motion are the following:

$$F(\mathbf{x_t}) = \mathbf{x_t}; \quad G(\mathbf{x_t}) = \begin{pmatrix} \cos\theta_t & 0 \\ \sin\theta_t & 0 \\ 0 & 1 \end{pmatrix} \tag{2}$$

Thus, the equation of the motion can be written as:

$$\begin{pmatrix} p_{x,t+1} \\ p_{y,t+1} \\ \theta_{t+1} \end{pmatrix} = \begin{pmatrix} p_{x,t} \\ p_{y,t} \\ \theta_t \end{pmatrix} + \begin{pmatrix} \cos\theta_t & 0 \\ \sin\theta_t & 0 \\ 0 & 1 \end{pmatrix} \begin{pmatrix} v_t \\ w_t \end{pmatrix} \tag{3}$$

At each time step, each agent chooses a control input that brings it to the next location (which can be different from the current one or the same as the current one).

Let $sh^i(\mathbf{x})$ be the mapping between the agent's state $\mathbf{x}$ and the set of points in $\mathbb{R}^2$ occupied by the agent in this state. In other words, $sh$ defines the footprint of the agent when it is located in a specific state.

A control sequence (or just control) for an agent $i$ is a mapping $U^i : T \to \mathbb{R}^m$, that can be written as $U^i = \{\mathbf{u}_0^i, \mathbf{u}_1^i, \mathbf{u}_2^i, \ldots\}$. The application of controls $U^i$ to system 1 produces a trajectory (or path) in the workspace. Path can be written as a mapping $\pi^i : T \to \mathbb{R}^n$ or as a sequence of states in discrete time $\pi^i = \{\mathbf{x}_0^i, \mathbf{x}_1^i, \mathbf{x}_2^i, \ldots\}$.

In this work, we are interested in converging solutions, *i.e.*, controls by which an agent reaches the neighborhood of a particular location and never moves away from it. Further on, we will use the terms path and trajectory interchangeably.

The time step, $t_{fin}^i \in T$, by which the agent $i$ reaches the neighborhood of its final destination defines the duration of the trajectory: *duration* $(\pi^i) = t_{fin}^i$.

The control sequence for the agent $i$ is valid w.r.t. *iff* at every step $t$ control $U^i(t) = \mathbf{u}_t^i$ satisfy the control constraints:

$$\forall i \in \mathcal{N} \ \forall t \in [0, \ t_{fin}^i - 1] \ \forall k \in [1, \ m], \ u_{k,min} \le U^i(t)_k \le u_{k,\,max} \tag{4}$$

Consider now the two control sequences for distinct agents: $U^i, U^j$. Let $\mathbf{x}_t^i$ and $\mathbf{x}_t^j$ be the states of the agents $i$ and $j$ at the time $t$ when executing the controls $U^i, \ U^j$ using the motion model in Eq. (1). Trajectories are called conflict-free if the agents following them never collide, that is:

$$sh(\mathbf{x}_t^i) \cap sh(\mathbf{x}_t^j) = \varnothing, \ \forall t \in [0, \ max \ (t_{fin}^i, \ t_{fin}^j)]$$

We assume that the $\Delta t$ value is small enough, and if two agents $i$ and $j$ are collision-free at time moments $t$ and $t + \Delta t$, then they also did not collide when moving between corresponding states $\mathbf{x}_t^i, \ \mathbf{x}_t^j$ and $\mathbf{x}_{t+\Delta t}^i, \ \mathbf{x}_{t+\Delta t}^j$.

**The problem** now is to find a control sequence for each agent $i$ s.t. *(i)* each individual control sequence is valid; *(ii)* the produced trajectory starts at the predefined initial state and ends in the neighborhood $\mathscr{B}_\varepsilon(\mathbf{g}^i)$ of the corresponding goal $\mathbf{g}^i \in G$; *(iii)* each pair of corresponding trajectories is conflict-free, *i.e.*, the agents following them never collide.

If now the set of control sequences (one for each agent is given) is presented as a solution, its cost can be defined using produced trajectories $\pi^i$ as follows:

$$makespan = \max \ (duration \ (\pi^i)) \tag{5}$$

We are not imposing a strict requirement on optimizing the cost of the solution, but lower-cost solutions are obviously preferable.

## Additional assumptions

In the rest of the article, we rely on the following assumptions, which are not uncommon in the field: *(i)* all agents are homogeneous, and every agent $i$ is represented as an open disk of radius $r^i \in \mathbb{R}^+$; *(ii)* each action is executed perfectly, so at each time step, an agent's position is deterministic and known exactly.

Based on the first assumption, agents $i$ and $j$ are in a collision-free state if the distance between them is greater (or equal) than the sum of their radii:

$$sh(\mathbf{x_t^i}) \cap sh(\mathbf{x_t^j}) = \varnothing \iff ||\mathbf{x_t^i} - \mathbf{x_t^j}||_2 \geq r^i + r^j \tag{6}$$

where $|| \cdot ||_2$ is Euclidean vector norm.

In this work, we consider a decentralized setting, which implies that each agent acts independently and makes decisions on what controls to choose based on limited information gained from local observation/communication. In particular, we adopt the following assumptions regarding this aspect of the problem.

We assume that at each time step $t \in T$, each agent $i \in \mathcal{N}$, indeed, knows its own state $\mathbf{x}_t^i$, velocity $\mathbf{v}_t^i = (\frac{p_{x,t} - p_{x,t-1}}{\Delta t}, \frac{p_{y,t} - p_{y,t-1}}{\Delta t})$ and radius $r^i$. At the same time, communication with other agents is not assumed, the agents can only use outer information on neighboring agents that can be observed, namely, current positions, current velocities and radii. In other words, the inner agent state (*e.g.*, preferred velocity or current selected control) is not available for other agents. Moreover, the visibility range is limited, and only the agents within the predefined visibility range $R^i \in \mathbb{R}^+$ are observable. No communication between agents is assumed.

## BACKGROUND

The proposed approach relies on the concepts of the *Model Predictive Path Integral* algorithm. The *Model Predictive Path Integral* (or *MPPI*) algorithm was developed to solve the nonlinear stochastic optimal control problem. The *MPPI* algorithm adopts the control scheme from the *MPC* methods, where the problem is solved incrementally, and at each step, a control sequence is generated for a small time horizon. The primary distinction from *MPC* is that optimal control is constructed using a sampling-based optimization method. At each time step, *MPPI* samples a set of control inputs and generates a set of corresponding trajectories using a given model of the dynamic system. Each trajectory is evaluated using a cost function, and based on the resulting costs, the control input is selected. Therefore, the algorithm does not rely on linear and quadratic approximations of dynamics and cost functions, enabling its application to a wide range of tasks (*Williams et al., 2016*). Additionally, each sequence (and the corresponding trajectory) can be sampled and evaluated independently, allowing for parallelization to enhance performance (*Williams, Aldrich & Theodorou, 2017*).

Moreover, for sampling safe control actions, it is essential to incorporate control constraints that ensure the safety of future movements. The *Optimal Reciprocal Collision Avoidance* (*ORCA*) algorithm was selected for this purpose. It is based on the concept of identifying a set of relative velocities that could result in a collision with a neighboring

agent. Using this set, a linear constraint is formulated to establish a safe region in the velocity space. While this algorithm offers sufficient conditions for safe motion, it is also less prone to deadlocks or oscillations compared to, for example, *BVC*. Let us delve into these approaches in more detail.

## MODEL PREDICTIVE PATH INTEGRAL

Originally presented in *Williams et al. (2016)*, the *MPPI* algorithm relies on the assumption that the dynamical system is represented in a nonlinear affine form. The article by *Williams et al. (2017)* discusses the evolution of the original idea, introducing the *Information Theoretic Model Predictive Control* (*IT-MPC*) algorithm. This algorithm allows for the consideration of a broader class of dynamical systems given in a general nonlinear form. Subsequent research has focused on enhancing the robustness and sampling efficiency of the algorithm, as indicated by works such as those by *Gandhi et al. (2021)*, *Balci et al. (2022)* and *Tao et al. (2022a, 2022b)*. For instance, the works by *Tao et al. (2022a, 2022b)* explore enhancing safety by exploring new sampling parameters considering control barrier function. Another area of development aims at obtaining smoother trajectories, as highlighted in the work by *Kim et al. (2022)*. Additionally, there are studies dedicated to adapting the *MPPI* algorithm to a multi-agent formulation. Such approaches often assume communication between agents (*Wang et al., 2022*; *Song et al., 2023*) or rely on predicting the motion of other agents without guaranteeing the safety of the resulting actions (*Streichenberg et al., 2023*). A more detailed description of the *MPPI* algorithm (hereafter referred to as the *IT-MPC* version) will be presented. The stages of the algorithm are illustrated in Fig. 2.

The problem of stochastic optimal control is under consideration where the works dedicated to *MPPI* present the stochastic dynamical system equation of motion in a general form:

$$\mathbf{x_{t+1}} = \mathscr{F}(\mathbf{x_t}, \ \tilde{\mathbf{u}}_\mathbf{t}) \tag{7}$$

where $\mathbf{x_t} \in \mathbb{R}^n$ is the state of the system at time $t$, $\tilde{\mathbf{u}}_\mathbf{t} \in \mathbb{R}^m$ is the input of the system at time $t$. $\mathscr{F} : \mathbb{R}^n \times \mathbb{R}^m \to \mathbb{R}^n$ is a time-invariant (generally non-linear) state-transition function, and there is a time horizon $t \in \{0, 1, 2, \ldots, T\}$. It is assumed that there is no direct control over the input $\tilde{\mathbf{u}}_\mathbf{t}$, but $\tilde{\mathbf{u}}_\mathbf{t} \sim \mathscr{N}(\mathbf{u_t}, \ \mathbf{\Sigma})$ is a random vector, and there is a direct control over the mean $\mathbf{u_t} \in \mathbb{R}^m$. Note, that Eq. (1) (that is a part of the definition of the problem considered in this article) is a special case of Eq. (7).

Let us define $U = (\mathbf{u_0}, \ \mathbf{u_1}, \ \ldots, \mathbf{u_{T-1}})$ as the control sequence, $\mathscr{U}$ is the set of admissible control sequences, $\tilde{U} = (\tilde{\mathbf{u}}_\mathbf{0}, \ \tilde{\mathbf{u}}_\mathbf{1}, \ \ldots, \ \tilde{\mathbf{u}}_{\mathbf{T-1}})$ as input sequence and $X = (\mathbf{x_0}, \ \mathbf{x_1}, \ \ldots, \mathbf{x_T})$ as the state trajectory over the time horizon $T$. Let us denote the distribution of overall input sequence $\tilde{U}$ as $\mathbb{Q}$. The optimal control problem is defined as:

$$U^* = \underset{U \in \mathscr{U}}{\arg \min} \ \mathbb{E}_{\mathbb{Q}} \left[ \mathscr{L}(X, U) \right], \tag{8}$$

where $\mathscr{L}$ is a specifically-designed cost function. In other words, the task is to find such a sequence of controls for which the expected value of the cost function $\mathscr{L}$ with respect to the probability measure $\mathbb{Q}$ induced by the controlled dynamics is minimal.

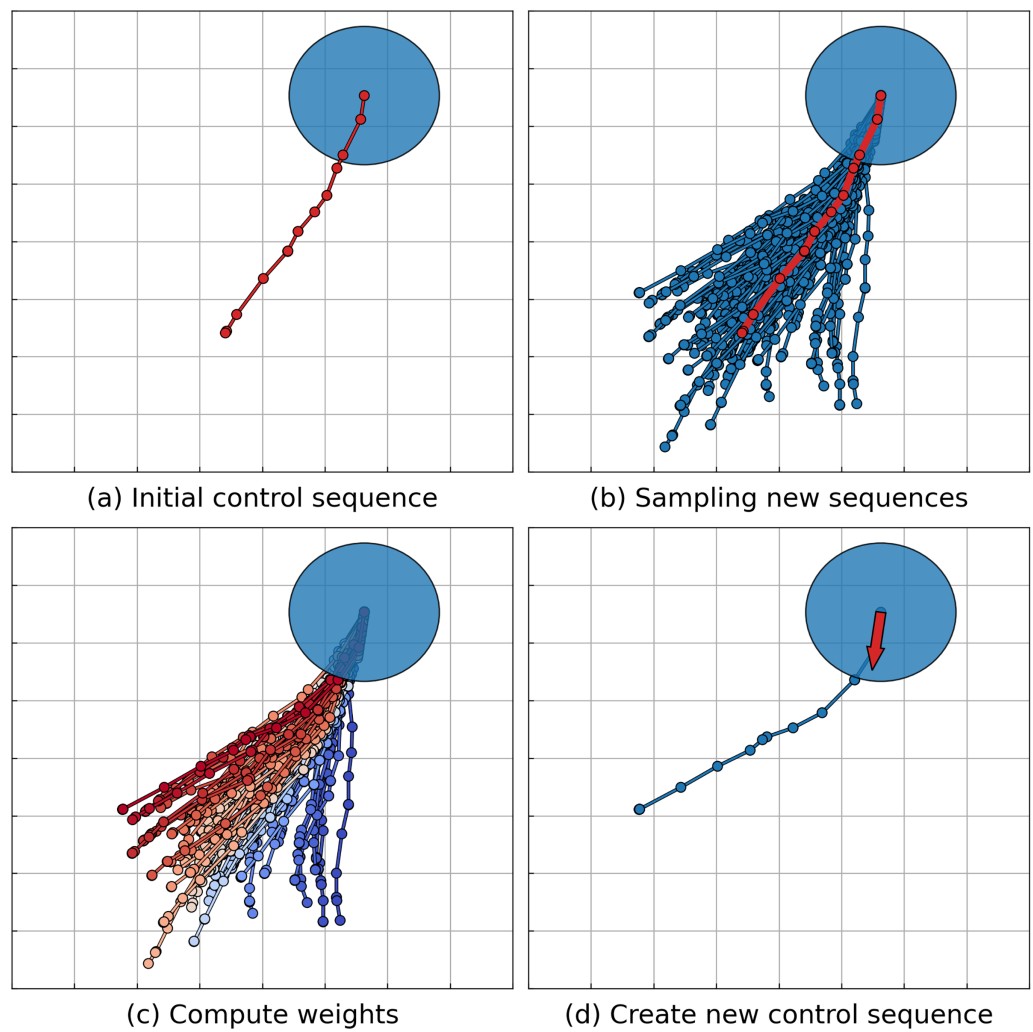

(a) Initial control sequence

(b) Sampling new sequences

(c) Compute weights

(d) Create new control sequence

**Figure 2 Illustration of the MPPI algorithm step.** The agent's footprint is depicted as a blue disk. (A) The algorithm retrieves the solution (control sequence) from the previous step to serve as a foundation for further sampling. The corresponding trajectory is represented by a red line. (B) A set of control sequences is generated through sampling, with corresponding trajectories depicted as blue lines. The mean value of the sampling distribution is the control sequence from the previous step, with its corresponding trajectory shown as a red line. (C) Each control sequence undergoes evaluation and is assigned a weight based on the evaluation. Trajectories with higher weights are highlighted in red, while those with lower weights are marked in blue. (D) The new solution, represented as a blue line, is derived as a weighted sum of the sampled sequences. Agents execute only the first control from the sequence (shown as a red arrow) and repeat the sampling process.

The *MPPI* family of the algorithms consider the cost functions of the form:

$$\mathcal{L}(X, U) = \phi(\mathbf{x_t}) + \sum_{t=0}^{T-1} (r(\mathbf{x_t}) + c(\mathbf{u_t})), \qquad (9)$$

where $\phi(\mathbf{x_t})$ is terminal cost function, $r(\mathbf{x_t})$ is running cost function and $c(\mathbf{u_t})$ is running control cost function. For example, in the single-robot navigation task, one of the options to define the terminal cost-function $\phi(\mathbf{x_t})$ might be to compute the distance between the

last point of the trajectory $\mathbf{x_t}$ and the target position. Cost term $r(\mathbf{x_t})$ might measure the distance from the robot to the obstacles or to some reference geometric path at each point of the trajectory $\mathbf{x_t}$.

The *MPPI* algorithms work as follows. Let the current state $\mathbf{x}_0$ and some initial sequence of controls $U^{init} = (\mathbf{u_0^{init}}, \ldots, \mathbf{u_{T-1}^{init}})$ be given (a trajectory corresponding to the initial control sequence is depicted on Fig. 2A as the red line). First, the *MPPI* algorithm samples a set of $K$ sequences:

$$\xi^k = (\boldsymbol{\varepsilon_0^k}, \boldsymbol{\varepsilon_1^k}, \ldots, \boldsymbol{\varepsilon_{T-1}^k}), \ \boldsymbol{\varepsilon_t^k} \sim \mathcal{N}(\mathbf{0}, \Sigma). \tag{10}$$

Based on $U^{init}$ and $\xi^k$, a set of $K$ control sequences $U^k$ is obtained:

$$U^k = (\mathbf{u_0^k}, \mathbf{u_1^k}, \ldots, \mathbf{u_{T-1}^k}), \ \mathbf{u_t^k} = \mathbf{u_t^{init}} + \boldsymbol{\varepsilon_t^k}. \tag{11}$$

Using $x_0$ and the equation of motion 7, a corresponding trajectory $X^k$ is obtained for every sequence of controls $U^k$ (a set of such trajectories is shown on Fig. 2B as blue lines, red line represents the initial trajectory $U^{init}$):

$$X^k = (\mathbf{x_0}, \ F(\mathbf{x_0}, \mathbf{u_0^k}), \ F(\mathbf{x_1^k}, \mathbf{u_1^k}), \ \ldots, \ F(\mathbf{x_{T-1}^k}, \mathbf{u_{T-1}^k})). \tag{12}$$

Each sequence of controls is evaluated, and its cost is computed. These costs are used to calculate the weights $\omega(U)$ for importance sampling and obtain the resulting sequence of controls $U^* = (\mathbf{u_0^*}, \mathbf{u_1^*}, \ldots, \mathbf{u_{T-1}^*})$[1].

$$\mathbf{u_t^*} = \sum_{k=1}^{K} \left( \omega(U^k) \mathbf{u_t^k} \right). \tag{13}$$

After the resulting controls are computed, control $\mathbf{u_0^*}$ is executed and new initial controls sequence $U^{init}$ is built on the basis of the $U^*$ by removing $\mathbf{u_0^*}$ and adding some default initial control value $\mathbf{u^{init}}$ to the end (a trajectory corresponding to resulting control sequence depicted on Fig. 2D as blue line, executed control is shown as red arrow):
$$U^{init} = (\mathbf{u_1^*}, \mathbf{u_2^*}, \ldots, \mathbf{u_{T-1}^*}, \mathbf{u^{init}})$$

Thus, the *MPPI* algorithm allows solving nonlinear stochastic optimal control problems for a single agent. But in a decentralized multi-agent case it can create unsafe solutions (if all sampled trajectories will lead to collisions and have a lot of weight), or require numerous samples to find a safe solution.

## OPTIMAL RECIPROCAL COLLISION AVOIDANCE

The *Optimal Reciprocal Collision Avoidance* (*ORCA*) method (*Van Den Berg et al., 2011a*) represents a development of the theory of velocity obstacles (*Fiorini & Shiller, 1998*). The *ORCA* algorithm is designed to address the issue of decentralized reciprocal multi-agent collision avoidance. In essence, the method identifies a velocity that will not result in a collision with other agents, solely based on observable information and the assumption that other agents are following the same policy.

[1] Please note that the cost function that is used to evaluate the constructed trajectories, $S(X, U)$ may be different from the one used in defining the *MPPI* control problem, $\mathcal{L}(X, U)$. Finding such $S(X, U)$ that the resultant $U^*$ is optimal in terms of the expected value of Eq. (9), is one of the key issues considered in the works devoted to IT-MPC approach. The authors of *MPPI* propose several cost functions $S(X, U)$ and we refer the reader to *Williams et al. (2017)* and *Williams (2019)* for more details.

The method employs an iterative scheme to select a safe velocity. The solution is obtained as follows: the agent updates its observations and information regarding its state. Subsequently, in the velocity space, an area of safe actions bounded by linear constraints is established. Within this area, a search for a velocity that satisfies all linear constraints and is closest to a preferred speed is conducted using a linear programming algorithm. Following this, the selected velocity is executed, and the agent's state is updated.

Let us explore the process of constructing linear constraints for a scenario involving two agents. (For a greater number of agents, the same procedure is repeated for each visible agent.) Let $\mathbf{v_t^i}$ be a change of the agent's $i$ position per unit of time $\Delta t$.

$$\mathbf{v_t^i} = \left( \frac{p_{x,t} - p_{x,t-1}}{\Delta t}, \frac{p_{y,t} - p_{y,t-1}}{\Delta t} \right). \tag{14}$$

For the sake of simplicity, we will not specify the index $t$, all further constructions will be made for time $t$. Firstly, the agent $i$ constructs a set of relative velocities $VO_{i|j}^\tau$ (velocity obstacle) that will result in collision between agents $i$ and $j$ at some moment before time $t + \tau$.

$$VO_{i|j}^\tau = \{\mathbf{v_{i|j}} \mid \exists t' \in [t, t + \tau] : (t' - t) \cdot \mathbf{v_{i|j}} \in D(p_x^j - p_x^i, p_y^j - p_y^i, r^i + r^j)\}, \tag{15}$$

where $(p_x^i, p_y^i)$, $(p_x^j, p_y^j)$ are positions of agents $i$ and $j$, $r^i$, $r^j$ are radii of agents $i$ and $j$ and $D(p_x, p_y, r)$ is an open disk with center in point $(p_x, p_y)$ and radius $r$.

On the basis of the $VO_{i|j}^\tau$ a linear constraint $ORCA_{i|j}^\tau$ is constructed as follows. Assume that the velocities $\mathbf{v^i}$, $\mathbf{v^j}$ will lead the agents to a collision at time, $t' : (t' - t) < \tau$ i.e., $\mathbf{v_{i|j}^{rel}} = (\mathbf{v^i} - \mathbf{v^j}) \in VO_{i|j}^\tau$. Let $\mathbf{u}$ be a vector to the nearest point on the boundary of $VO_{i|j}^\tau$ from $\mathbf{v_{i|j}^{rel}}$. In other words, $\mathbf{u}$ can be interpreted as the smallest change in relative velocity $\mathbf{v_{i,j}^{rel}}$ to prevent a collision within time $\tau$.

$$\mathbf{u} = \left( \arg\min_{\mathbf{v} \in \partial VO_{i|j}^\tau} \|\mathbf{v} - \mathbf{v_{i,j}^{rel}}\| \right) - \mathbf{v_{i,j}^{rel}}. \tag{16}$$

Let $n$ be the vector normal of the boundary of $VO_{i|j}^\tau$ at point $\mathbf{v_{i,j}^{rel}} + \mathbf{u}$ and directed outward to the boundary. Then, the set of collision-free velocities with respect to the agent $j$ can be constructed as a half-plane bounded by a line passing through the point $\mathbf{v_i} + \alpha\mathbf{u}$ and perpendicular to the vector $\mathbf{n}$.

$$ORCA_{i|j}^\tau = \{\mathbf{v} \mid (\mathbf{v} - (\mathbf{v_i} + \alpha_{resp}\mathbf{u})) \cdot \mathbf{n} \geq 0\}, \tag{17}$$

where $\alpha_{resp}$ is responsibility factor. For agents avoiding collisions reciprocally, it should be set to be greater or equal to $1/2$. When avoiding collisions with static or dynamic obstacles, it should be set $\alpha_{resp} = 1$.

The half-plane $ORCA_{j|i}^\tau$ for agent $i$ is defined symmetrically. The described algorithm also applied if $i$ and $j$ are not on a collision course.

Thus, the above algorithm effectively enables the derivation of linear constraints for assessing the safety of velocities. Furthermore, the symmetrical construction of linear constraints in the velocity space enables agents to orient themselves such that, in their

efforts to avoid collisions, they move in different directions relative to each other (thereby reducing the likelihood of deadlock occurring). However, it is important to note that this method only allows for finding solutions in the velocity space, and the solutions obtained may not be feasible in the presence of kinematic constraints.

# MODEL PREDICTIVE PATH INTEGRAL FOR MULTI-AGENT COLLISION AVOIDANCE

## Preliminaries

A straightforward way to adapt the *MPPI* algorithm for collision avoidance is by adding a penalty to the cost function for the intersection or proximity of the sampled trajectories with other objects. In the case of multi-agent scenarios, it may be beneficial to utilize the planned trajectories of other agents to assess potential collisions. Consequently, those controls corresponding to more hazardous trajectories will carry less weight, and when averaged (see Eq. (13)), will exert a reduced influence on the final control. A more sophisticated approach (for example, *Streichenberg et al., 2023*) may involve filtering out dangerous trajectories from the overall sample, excluding control sequences that could lead to collisions from the weighted sum.

The primary challenge in a decentralized scenario is that agents lack access to information about the planned trajectories of other agents. This necessitates the prediction of trajectories based on available data. Several works focusing on classical *MPC* for multi-agent tasks address this problem (*Streichenberg et al., 2023*) (*e.g.*, *Zhu & Alonso-Mora, 2019*; *Cheng et al., 2017*). In this article, we consider a scenario in which only information on the positions and current velocities of other agents is available. Consequently, the prediction can be based on a constant velocity model, assuming that other agents will continue to move at their current velocities. However, it is important to note that more complex and sophisticated motion prediction models can be employed, such as those based on neural networks (*Gulzar, Muhammad & Muhammad, 2021*).

The basic approach to adapting *MPPI* for decentralized multi-agent scenarios can reduce the likelihood of collisions. However, since agents need to make decisions about their actions in parallel and independently, without the ability to communicate, the chosen actions cannot guarantee the safety of the found controls. For instance, consider the scenario from Fig. 3A. Two agents (yellow and green circles) attempt to predict trajectories (dotted arrows) of other agents (blue circles). They realize that their current controls could lead to collisions with other agents, so they opt for alternative "safe" actions (colored arrows in Fig. 3B). However, since the green and yellow agents made decisions in parallel and independently, a collision occurs between them (Fig. 3C). Additionally, when excluding unsafe control sequences from the sample, it may be necessary to continue sampling until the number of safe trajectories reaches a predefined minimum. This, in turn, could significantly reduce the algorithm's performance.

To ensure collision-free motion in a decentralized setting, it is essential for all agents to adhere to a common set of rules, expressed as a set of constraints built according to the same principle. An example of such principles can be the *ORCA* algorithm discussed earlier. A direct method to incorporate various constraints in stochastic optimization is to

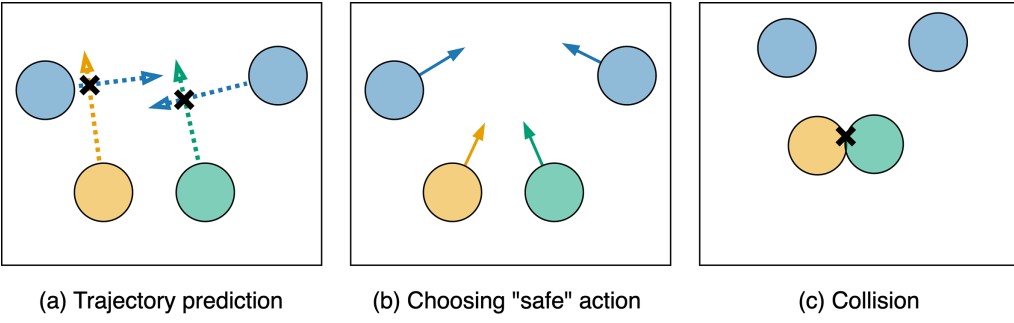

(a) Trajectory prediction  (b) Choosing "safe" action  (c) Collision

**Figure 3** An example of a scenario where choosing an action based on predicted trajectories can lead to a collision between agents.  

significantly reduce the weight or exclude from the weighted sum those elements of the sample that do not satisfy these constraints. However, this approach noticeably reduces the sampling efficiency, as constraints can eliminate a significant portion of the sample elements.

Consider the example in Fig. 1. The brown agent is moving toward its goal but is surrounded by blue agents preventing its movement (Fig. 1A). Based on available information, *ORCA* constraints were constructed in the velocity domain (Fig. 1B, with the white area representing a set of safe velocities). Next, the agent samples a batch of controls using a normal distribution with some initial parameters (the corresponding velocities are shown as brown dots on Fig. 1B, and the brown circle represents the confidence interval of $4\sigma$), and it is evident that a significant number of samples lie outside the safe zone.

As a result, it is necessary to significantly reduce the number of samples that do not satisfy the set of constraints. To address this, we propose finding distribution parameters close to the original ones, such that the sampled controls satisfy the constraints with a predefined probability (a similar technique was also described in *Tao et al. (2022a, 2022b)*, but the described approach only considers the single-agent case and does not guarantee collision-free motion). Additionally, control limits can be included in this problem as constraints, which can also improve sampling efficiency. In the following sections, we will demonstrate that the search for such parameters can be reduced to a convex optimization problem. An illustration of sampling from a safe distribution is shown in Fig. 1C (with green dots representing velocities corresponding to the sampled controls, the green ellipse representing the confidence interval of $4\sigma$, and the orange area illustrating control limits constraints).

Thus, the final pipeline of the proposed algorithm is as follows (Fig. 4). Firstly, the algorithm constructs a set of linear constraints for the current state. After that, the safe distribution parameters should be determined. Furthermore, based on the available information, the trajectories of other agents should be predicted. Next, a set of control sequences is sampled. Moreover, the first control in each set is sampled using the safe distribution. This ensures that the next selected control will not lead to a collision. Each control sequence is evaluated, including penalties for conflicts with the predicted trajectories of other agents. Finally, based on the estimates obtained in the previous step, a

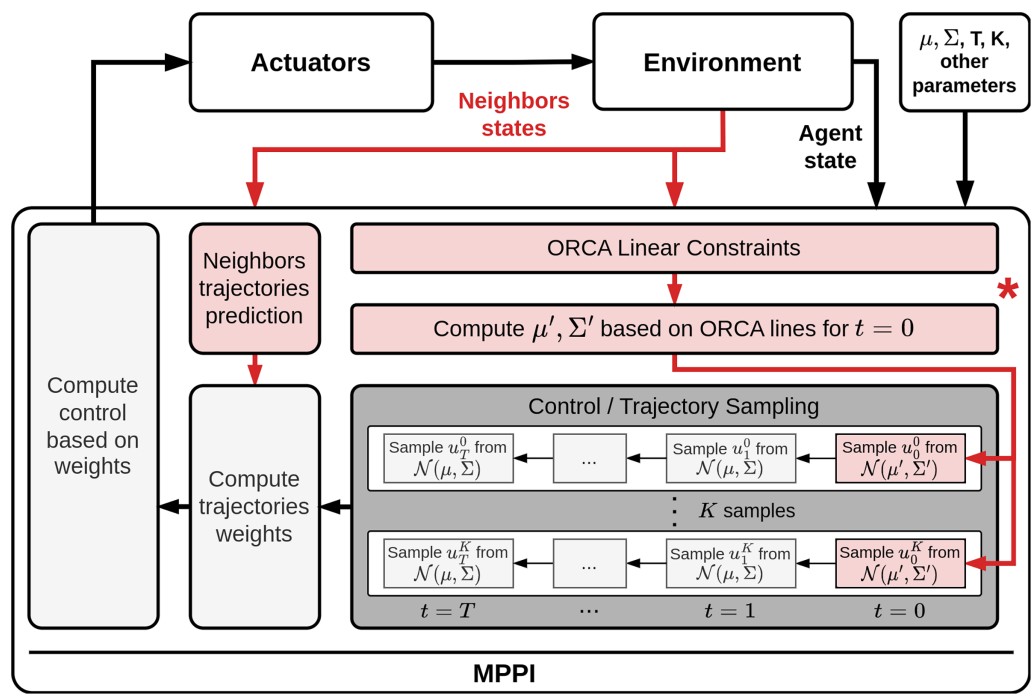

**Figure 4 Common collision avoidance pipeline based on MPPI algorithm.** The red elements on scheme demonstrate the components that were added to MPPI procedure to adapt in for decentralized multi-agent scenario. (*) The main contribution of our work is introducing of a method for taking into account linear constraints in the sampling process. The corresponding component is marked with an asterisk on the scheme.

control action is formulated and executed. For a more detailed description of the algorithm, we refer the reader to the Supplemental Materials, where a detailed pseudocode of the algorithm routine is presented.

The most significant stage of the suggested approach is the construction of a new distribution for sampling safe control actions (the corresponding component is marked with an asterisk on the scheme in Fig. 4). In the subsequent sections, a comprehensive exposition of this procedure will be presented, along with a technique for reducing this task to second-order cone programming (SOCP). Additionally, a theoretical examination of the proposed approach will be provided.

## Constructing safe distribution

As mentioned earlier, to incorporate collision avoidance linear constraints into the sampling process, it is essential to determine new distribution parameters that are close to the initial ones $\mu$, $\Sigma$. The objective is to ensure that the probability of the constraint being satisfied by the velocity vector obtained based on the sampled control is higher than a certain predefined value $\alpha$, which is close to 1. This task can be formulated as an optimization problem using the chance constraints representation of linear constraints.

First, we need to define the closeness of new distribution parameters $\mu'$, $\Sigma'$ for $\mathbf{u_t} = \{u_{t,1}, \ldots, u_{t,m}\}$ with the initial $\mu$, $\Sigma$. In this article, we assume that components of $\mathbf{u_t}$

are independently normally distributed. Then, it can be done using 1-norm $||\mu' - \mu||_1$ and $||diag\,(\Sigma') - diag\,(\Sigma)||_1$, where $diag(A)$ is a vector constructed from main diagonal elements of some matrix $A$. So, the objective function of optimization problem can be formulated in the next form:

$$||\mu' - \mu||_1 + ||diag\,(\Sigma') - diag\,(\Sigma)||_1. \tag{18}$$

Moreover, it should be noted, that elements of $diag\,(\Sigma') = (\sigma_0'^2,\ \sigma_1'^2,\ ...,\sigma_m'^2)$ must be positive or zero. So, additional constraints of the next form appear.

$$\sigma_k'^2 \geq 0,\quad k = 1,\ ...m. \tag{19}$$

Next, we need to consider collision avoidance constraints. In our work we suggest using velocity constraint from *ORCA* algorithm, but other linear constraints on velocity (*e.g.*, from *Velocity Obstacles* based methods) or position (*e.g.*, from *BVC* based methods) can be applied in the similar manner.

Let $\mathcal{N}_i'$ be a set of visible neighbors of some agent $i$. Let us denote single linear constraint $ORCA_{i|j}^{\tau}$ for agent $i$ and other agent $j \in \mathcal{N}_i'$ as triple $(a_j,\ b_j,\ c_j)$. So, velocity vector $\mathbf{v_t^i} = (v_{x,t}^i,\ v_{y,t}^j)$ satisfies it if the following inequality is true:

$$a_j \cdot v_{x,t}^i + b_j \cdot v_{y,t}^j + c_j \leq 0. \tag{20}$$

The velocity vector (*i.e.*, movement per unit of time) for a given state $\mathbf{x_t^i}$ and control action $\mathbf{u_t^i}$ can be predicted using the model from Eq. (1). Since the state $\mathbf{x_t}$ includes the agent's position $(p_{x,t},\ p_{y,t})$, the velocity vector can be rewritten so that linearly depends on the control $\mathbf{u_t^i}$ (for a fixed state $\mathbf{x_t^i}$, the $F(\mathbf{x_t})$ and $G(\mathbf{x_t})$ values can be considered as a constant vector $\mathbf{F}$ and matrix $G$).

$$\begin{aligned}v_{x,t}^i &= F_1 + G_{1,1}u_{t,1} + ... + G_{1,m}\,u_{t,m} - p_{x,t}\\ v_{y,t}^i &= F_2 + G_{2,1}u_{t,1} + ... + G_{2,m}\,u_{t,m} - p_{y,t}.\end{aligned} \tag{21}$$

Let $\mathbf{a_j'} = \{(a_jG_{1,1} + b_jG_{2,1}),\ ...,(a_jG_{1,m} + b_jG_{2,m})\}$ and $b_j' = -(c_j + a_j(F_1 - p_{x,t}) + b_j(F_2 - p_{y,t}))$. So, the inequality 20 can be rewritten in the following form

$$\mathbf{a_j'}T\mathbf{u_t} \leq b_j'. \tag{22}$$

While sampling, we can consider $\mathbf{u_t} = \{u_{t,1},\ ...,u_{t,m}\}$ as stochastic vector with independently normally distributed components. Then, involving chance constraints theory (for more information see Theorem 4.9 in Section 4.3 "*Chance-Constrained Programming*" from *Liu & Liu (2009)*), $Pr\{\mathbf{a_j'u_t} \leq b_j'\} \geq \alpha$ *if and only if*:

$$\mathbf{a_j'}T\mu' + \Phi^{-1}(\alpha)\sqrt{\mathbf{a_j'}T\Sigma'\mathbf{a_j'}} \leq b_j', \tag{23}$$

where $\mu', \Sigma'$ are mean and covariance matrix of $\mathbf{u_t}$, $\Phi(\cdot)$ is the standard normal cumulative distribution function and $\alpha$ is the required confidence level. We also

can incorporate controls limitations from Eq. (4), which can be rewritten using chance-constraints:

$$
\begin{aligned}
\mu_k{}' + \Phi^{-1}(\alpha)\sigma_k' &\leq u_{k,max}, \\
\mu_k{}' - \Phi^{-1}(\alpha)\sigma_k' &\geq u_{k,min}, \\
k = 1, &\ldots, m,
\end{aligned}
\tag{24}
$$

where $\mu_k{}'$ is component of mean $\mu'$, $\sigma_k'$ is element from main diagonal of $\Sigma'$.

So, the optimization problem to create new distribution parameters for agent $i$ can be written as:

$$
\begin{aligned}
\arg\min_{\mu', \Sigma'} \quad & ||\mu' - \mu||_1 + ||diag(\Sigma') - diag(\Sigma)||_1 \\
\text{s.t.} \quad & \mathbf{a_j'}T\mu' + \Phi^{-1}(\alpha)\sqrt{\mathbf{a_j'}\Sigma'\mathbf{a_j'T}} \leq b_j', \ \forall j \in \mathcal{N}_i' \\
& \mu_k{}' + \Phi^{-1}(\alpha)\sigma_k' \leq u_{k,max}, \ k = 1, \ldots, m \\
& \mu_k{}' - \Phi^{-1}(\alpha)\sigma_k' \geq u_{k,min}, \ k = 1, \ldots, m \\
& \sigma_k' \geq 0, \ k = 1, \ldots, m.
\end{aligned}
\tag{25}
$$

Therefore, by solving the defined optimization problem, it is possible to derive new parameters for the sampling distribution so that, with a given probability $\alpha$, the sampled controls will comply with both safety constraints and control limits. In this section, we only address the general formulation of the optimization problem.

## Reducing to second-order cone program

As showed earlier, the problem of finding safe distribution parameters can be formulated as an optimization problem. Particularly, the problem Eq. (25) is a special case of a nonlinear programming problem. Solving the latter in the most general form is known to be NP-hard (*Murty & Kabadi, 1985*). However, numerous special cases exist (depending on how the objective function and the inequality constraints are formulated), that can be solved in polynomial time (*Nemirovski, 2004*).

In this section, we will demonstrate that the considered formulation of the optimization problem, *i.e.*, Eq. (25), can be rewritten as a Second-Order Cone Program (SOCP). This fact enables the use of a variety of approaches to find a solution for our problem, such as the interior-point method (*Kuo & Mittelmann, 2004*), which has polynomial time complexity. Additionally, there are numerous open-source and commercial solvers that handle such problem formulations (*e.g.*, the CVXPY Python library (*Diamond & Boyd, 2016*; *Agrawal et al., 2018*), the ECOS library (*Domahidi, Chu & Boyd, 2013*), *etc.*).

**Theorem 1.** *The optimization problem from Eq. (25) can be represented as a SOCP problem.*

*Proof.* The standard second-order cone program can be written as follows:

$$
\begin{aligned}
\text{minimize} \quad & \mathbf{f}^{Tx} \\
\text{s.t.} \quad & ||A_i\mathbf{x} + \mathbf{b_i}||_2 \leq \mathbf{c_i^T}\mathbf{x} + d_i, \ i = 1, \ldots, N,
\end{aligned}
\tag{26}
$$

where $\mathbf{x} \in \mathbb{R}^n$ is optimization variable, $\mathbf{f} \in \mathbb{R}^n$, $A_i \in \mathbb{R}^{n \times n_i}$, $\mathbf{b_i} \in \mathbb{R}^{n_i}$, $\mathbf{c_i} \in \mathbb{R}^n$ $d \in \mathcal{R}$ are problem parameters, $||\cdot||_2$ is Euclidean norm, $n_i$ is dimension of $i$ second-order cone constraint.

Firstly, we need to reformulate the objective function from Eq. (25) into a linear form, as shown in Eq. (26). To measure the distance between the new parameters and the initial ones, a 1-norm is employed. Hence, the objective function comprises a sum of absolute values. This type of function can be linearly represented by introducing auxiliary variables and additional linear constraints. So, our objective function can be represented in linear form, same as in SOCP.

Secondly, the linear constraint (for example, from Eqs. (19) and (24)) can be represented as second-order cone constraint with $n_i = 1$.

Finally, chance constraints Eq. (23) should be represented as second-order cone constraint. Let $\mathbf{x}$ be optimization variable, $A'_j \in \mathbb{R}^{2m \times 2m}$ is diagonal matrix and $\mathbf{c} \in \mathbb{R}^{2m}$ is a vector such that:

$$\mathbf{x} = (\mu'_1, \ldots, \mu'_m, \sigma'_1, \ldots, \sigma'_m, t_{\mu'_1}, \ldots, t_{\mu'_m}, t_{\sigma'_1}, \ldots, t_{\sigma'_m}) \tag{27}$$

$$A'_j = \begin{pmatrix} 0_{m,m} & I_m \mathbf{a}'_j & 0_{2m,2m} \end{pmatrix}, \tag{28}$$

$$\mathbf{c}'_j = -\frac{1}{\Phi^{-1}(\alpha)} \cdot (\mathbf{a}'_j, 0, \ldots, 0) \tag{29}$$

where $0_{m,m}$ is $m \times m$ zero matrix, $I_{n,m}$ is $n \times m$ zero matrix, $I_m$ is $m \times m$ identity matrix.

So, $\sqrt{\mathbf{a}'_j T \Sigma' \mathbf{a}'_j}$ can be rewritten as $||A'_j \mathbf{x}||_2$ and Eq. (23) can be represented in second-order form:

$$||A'_j \mathbf{x}||_2 \leq \mathbf{c}'_j \mathbf{T} \mathbf{x}' + \frac{1}{\Phi^{-1}(\alpha)} \cdot b'_j, \ \forall j \in \mathscr{N}'_i \tag{30}$$

Thus, the suggested optimization problem can be represented in the form of a second-order cone program.

Example detailed description of representation optimization problem in form of SOCP for single-integrator and differential-drive robot dynamics are presented in the Supplemental Materials.

## Control safety guarantee

In this section, we justify the safety guarantees of the suggested approach by presenting the following theorem.

**Theorem 2.** *Let i be the current agent with state* $\mathbf{x_t^j}$, $\mathscr{N}_i$ *is the set of neighboring agents with states* $\mathbf{x_t^j}, j \in \mathscr{N}_i$. *For any collision-free state* $\mathbf{x_t^j}$,

$$sh(\mathbf{x_t^i}) \cap sh(\mathbf{x_t^j}) = \varnothing, \ \forall j \in \mathscr{N}_i \tag{31}$$

*the control* $\mathbf{u_t^i}$ *selected by the suggested algorithm will not lead to a collision with probability* $\delta = \alpha^{|\mathscr{N}_i| \cdot K}$, *where K is a number of samples and $\alpha$ is a probability guarantee threshold.*

*Proof.* Let $ORCA_i^\tau = \{ORCA_{i|j}^\tau \mid j \in \mathscr{N}_i\}$ be the set of linear constraints that was obtained by ORCA algorithm based on available information about $\mathbf{x_t^i}$, $\mathbf{x_t^j}, j \in \mathscr{N}_i$. Using Eqs. (20)–(22) and controls limitation, linear constraints in control domain can be obtained. These constraints form a convex subset in control space $U_{safe}^i \subset \mathbb{R}^m$.

Let $\mu'$ and $\Sigma'$ are distribution parameters, obtained by solving optimization problem 25. According to Theorem 4.9 in Section 4.3 "Chance-Constrained Programming" from (*Liu & Liu, 2009*) variable $\mathbf{u}' \sim \mathcal{N}(\mu', \Sigma')$ satisfies $Pr\{\mathbf{a_j'u_t} \leq b_j'\} \geq \alpha$, for every single constraint. Thus, when sampling the value of $\mathbf{u}'$ from the distribution $\mathcal{N}(\mu', \Sigma')$, it will satisfy the constraint with $\alpha$ probability. So, $K$ samples satisfy $|\mathcal{N}_i|$ constraints with probability $\delta = \alpha^{|\mathcal{N}_i| \cdot K}$.

If all samples satisfy the constraints, then the weighted sum $\mathbf{u_t^*}$ (Eq. (13)) lies inside the convex subset $U_{safe}^i$. Therefore, the control $\mathbf{u_t^*}$ satisfies all the constraints, thus the corresponding velocity satisfies the *ORCA* constraints, which means the control $\mathbf{u_t^*}$ is safe (according to *Van Den Berg et al. (2011a)*).

**Remark 1.** *Note that the safety of the selected action can be guaranteed only if the visibility radius of the agent exceeds the maximum movement that the agent can make in one time step, otherwise it may encounter agents that lie outside the visibility range.*

**Remark 2.** *In order to guarantee safety when applying the proposed approach, it is necessary to select such parameters $\alpha$ and K that the $\delta$ probability does not exceed some desired threshold. For example, if the number of neighboring agents that are taken into account $|\mathcal{N}_i| = 5$ and we choosing the parameters $\alpha = 0.9999966$ $(\Phi^{-1}(\alpha) = 4.5)$, $K = 500$, then the probability $\delta = 0.991536009$.*

*However, high $\alpha$ values can lead to conservative solutions or deadlocks. This limitation can be avoided by using the idea of removing samples that lie outside the linear constraints. In this case, the sampling efficiency decreases slightly (for example, with $|\mathcal{N}_i| = 5, \alpha = 0.99865010$ $(\Phi^{-1}(\alpha) = 3.0)$ and $K = 500$, the probability $\delta$ will be 0.034, but only 3.37 samples will be removed in average), but strict guarantees of safety of the selected control will be given, since all elements of the weighted sum 13 will be guaranteed to lie inside the convex subset $U_{safe}^i$.*

*At the same time, the number of elements that would need to be removed from the sample when using the initial distribution parameters depends significantly on the current location of the linear constraints and may be a significant part. An example of such a sample is shown in Fig. 1B. So, all dots located outside the white area are potentially unsafe.*

**Remark 3.** *Note that in some cases, the problem may be infeasible if the set of safe velocities obtained from the ORCA algorithm is empty, or if there is no admissible control action that satisfies the constraints found. Possible solutions to this issue may include decelerating the agent or relaxing constraints by decreasing $\tau$ (refer to Theorem 4 in* Alonso-Mora et al. (2013)*). An approach similar to the one described in section "4.1.2 Double integrator dynamics" in* Zhu, Brito & Alonso-Mora (2022) *can also be utilized. It is based on increasing the radius of the agent taken into account by ORCA to obtain an additional buffer for braking.*

## EXPERIMENTAL EVALUATION

We implemented the proposed method in C++ (https://github.com/PathPlanning/MPPI-Collision-Avoidance) and evaluated its efficiency in various scenarios, including several dynamics. Firstly, we conducted experiments using a prevalent model of a differential drive robot's movement. The corresponding system model is widely discussed in works on

collision avoidance with kinematic constraints (*Snape et al., 2011*, *2010*; *Zhu, Brito & Alonso-Mora, 2022*), enabling us to compare it with several well-known methods. Subsequently, experiments were performed with a more complex car-like dynamic model. The third experiment aimed to compare the proposed approach with the existing state-of-the-art learning-based method. It is important to note that in all the launches, *(i)* all agents (robots) were simulated as disks, *(ii)* all agents (robots) had accurate information about their states, *(iii)* the controls were executed perfectly, and *(iv)* the movement was simulated according to the model described in the corresponding section. Lastly, we conducted an experiment involving heterogeneous agents (*i.e.*, agents with different sizes and speeds) and uncertainty in the available state information.

## Differential-drive dynamics

### Experimental setup

In the first series of experiments, the next model of differential-drive robot was used (a similar dynamic was described as an example in the problem statement section). The agent's state $\mathbf{x}$ was defined as $\mathbf{x} = (p_x, \ p_y, \ \theta)^T$, where $p_x, \ p_y$ is the position of the robot (center of corresponding disk) in 2D workspace, $\theta$–heading angle of the robot. The control was defined as $\mathbf{u} = (v, \ w)^T$, where $v$–linear velocity ($v_{min} \leq v \leq v_{max}$), $w$–angular velocity ($w_{min} \leq w \leq w_{max}$). The model involves next equations of motion:

$$\mathbf{x}_{t+1} = \mathbf{x}_t + \begin{pmatrix} \cos\theta_t & 0 \\ \sin\theta_t & 0 \\ 0 & 1 \end{pmatrix} \mathbf{u}_t \Delta t. \tag{32}$$

The following parameters for all robots were used in the experiments: robot sizes (radius of the corresponding disks) $r = 0.3 \ m$, linear velocity limits $v_{min} = -1.0 \ m/s$, $v_{max} = 1.0 \ m/s$, angular velocity limits $w_{min} = -2.0 \ rad/s$, $w_{max} = 2.0 \ rad/s$, sight/communication radius $R$ was not limited.

Three types of scenarios were used in the experiment: `Circle`, `Grid`, `Random`. Each scenario included several instances, which consists of $n$ pairs $(\mathbf{p}_{start}^i, \mathbf{g}^i)$, $i = 1, \ ..., n$ of initial states $\mathbf{p}_{start}^i = (p_{x,start}^i, p_{y,start}^i, \theta_{start}^i)$ and goal positions $\mathbf{g}^i = (p_{x,goal}^i, p_{y,goal}^i)$. The instance launch was considered successful if the algorithm found a collision-free solution that move every agent $i$ to some neighborhoods of the goal position $\mathscr{B}_\varepsilon(\mathbf{g}^i)$, and the number of simulation steps did not exceed some limit $k_{lim}$. The duration of the time step of simulation is fixed and equal to $\Delta t$. For all scenarios, the values of $\Delta t$, $\varepsilon$ and $k_{lim}$ were set to $0.1 \ s$, $0.3 \ m$ and $1,000 \ steps$, respectively. Each instance was launched multiple times.

**Circle scenario.** Let us consider the first experimental scenario. At the initial moment, a set of $n$ agents is located equidistant from each other on a circle with a diameter of $D_{circle} = 12 \ m$. The goal for each agent is to move to the position directly opposite their starting point. The initial heading angle is chosen so that the agent is oriented towards the goal position. The number of agents varies from 2 to 15 with an increment of 1. In this scenario, agents are generally sparsely distributed, but to reach their goals, they must pass through the center of the circle, which can lead to the formation of a dense cluster of agents in the center and potential deadlock situations. Each instance with a fixed number of

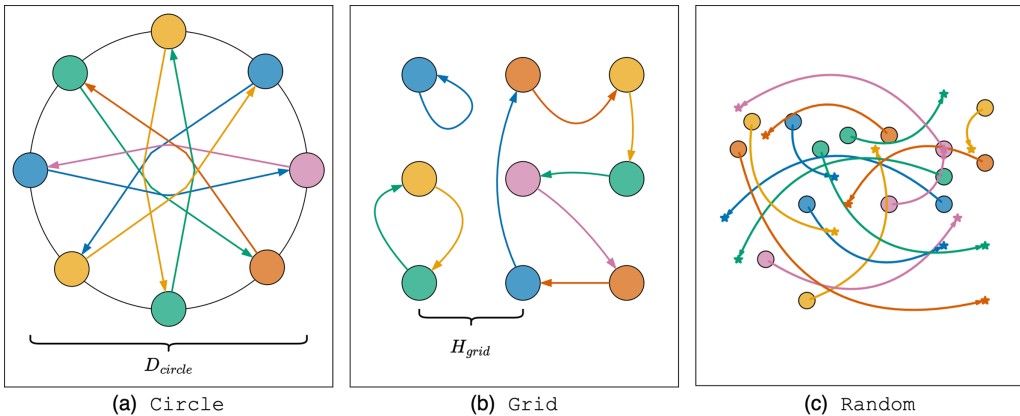

**(a)** Circle      **(b)** Grid      **(c)** Random

**Figure 5 Illustration of experimental scenarios with differential-drive robot kinematics.**

agents was launched 10 times. An illustration of the `Circle` scenario instance with eight agents is shown in Fig. 5A.

**Grid scenario.** In the second scenario, agents are located in the centers of cells of a regular square grid. Goal positions are obtained by randomly permutation agents between cells. The initial heading angle is selected equal to zero for all agents. The number of agents $n$ in the scenario varied between four ($2 \times 2$ grid), nine ($3 \times 3$ grid) and 16 ($4 \times 4$ grid). In addition, the size of the cell $H_{grid}$ varied, which allows changing the density of agents in the scenario. Three different cell sizes were selected for the experiment: $H_{grid} = 2.4 \ m = 8r$ (denote instances with such cell size as `sparse` instances), $H_{grid} = 1.8 \ m = 6r$ (denote instances with such cell size as `medium` instances), $H_{grid} = 1.5 \ m = 5r$ (denote instances with such cell size as `dense` instances). So, 10 instances were generated with a certain fixed number of agents $n$ and cell size $H_{grid}$ (90 instances in total) and each instance was launched 10 times. Such a scenario allows us to evaluate the behavior of algorithms when the agents are densely located. Note, however, that the cell sizes are chosen so that the distance between the agents is sufficient for movement even with doubling the radius with a safety buffer $2(r + \varepsilon_r)$ in algorithm ORCA-DD. An illustration of the scenario with $3 \times 3$ grid is presented in Fig. 5B.

**Random scenario.** The third scenario simulates the case of robots navigating in a common environment with a lower density and an irregular placement of start and goal positions. The scenario is constructed as follows. The area 20 $m \times$ 20 $m$ is discretized into a grid with a cell size of 1 $m \times$ 1 $m$ and 50 lists of 25 start/goal pairs are generated. The initial and goal position of each agent is selected as the center of a random cell inside the area. A cell cannot be selected by the start/goal of an agent if it is already selected by the start/goal of another agent in the current list or is a neighbor of another agent's start/goal cell. The initial heading angle is selected randomly. Instances are created by selecting the first $n$ agents from the same list, where $n$ varies from 5 to 25 in increments of 5 (250 instances in total, 50 for each number of agents). Each instance was launched 10 times. An illustration of the `Random` scenario instance with 15 agents is presented in Fig. 5C.

In addition to the proposed method, denoted as *MPPI*-ORCA, we also evaluate two collision avoidance algorithms: ORCA-DD (*Snape et al., 2010*) and B-UAVC (*Zhu, Brito & Alonso-Mora, 2022*). The first of them is based on the well-known ORCA (*Van Den Berg et al., 2011a*) algorithm, but uses a doubly increased radius of the agent for control, taking into account kinematic constraints. The second one is based on the Buffered Voronoi Cells approach, containing numerous improvements to the base method BVC, including consideration of kinematic constraints of the differential-drive robot. Both methods have been modified in such a way that a small random value $\varepsilon_x, \varepsilon_y \tilde{\mathcal{N}}$ is added to the goal direction vector. This was necessary to reduce the chance of getting into deadlocks in symmetric cases. In addition, for all the methods involved in the experiment, the radius of the agent used in the computations was increased by $\varepsilon_r = 0.05$ *m* relative to the real radius to minimize the chance of collision by creating additional safety buffer.

### Experimental results

**Success rate and makespan.** The main metrics measured during the experiment were *success rate* and *makespan*. *Success rate* is the percentage of successful instance launches. *Makespan* is the duration of the solution obtained at instance launch (*i.e.*, the time step when all agents have reached the target positions). Note that in all determined cases (*i.e.*, cases with perfect knowledge of agents' states) in our experiments, the *success rate* drop occurred due to going beyond the step limit, and not a single collision was observed in any algorithm.

As a result of the experiment on a `Circle` scenario, all launches of all algorithms were 100% successful. The average *makespan* and standard deviation for each number of agents in the circle are shown in Fig. 6. As you can see, the proposed algorithm on average provides better solutions than the other two evaluated approaches. And although the results of *MPPI*-ORCA and ORCA-DD partially overlap each other, however, the gap between them grows with an increase in the number of agents. At the same time, the results of algorithm B-UAVC are noticeably worse than those of *MPPI*-ORCA and ORCA-DD. For example, for 10 agents, the makespan of B-UAVC is on average 80% more than the same value of *MPPI*-ORCA.

When examining the trajectories in detail (illustration of solutions of an instance with 10 agents is given in Fig. 7), it can be observed that the ORCA-DD algorithm produces smoother trajectories. However, in some cases, the trajectories have a longer duration, and the speed of the agents is lower. This may be due to the use of a doubled radius in calculations, which, while reducing efficiency, also prevents the accumulation of a large number of agents in the center of the circle. Meanwhile, *MPPI*-ORCA, due to its stochastic nature, yields more fluctuating trajectories, but with lower durations for its solutions. Agents controlled by the B-UAVC algorithm move toward the goal until they approach each other at a small distance, leading to a dense accumulation of agents near the center of the circle and the formation of a deadlock. When resolving a deadlock, the algorithm generates trajectories with high cost.

The average *success rate* on `Grid`, grouped by the number of agents and the density of agents, is shown in Table 1. It is important to note that for each number of agents and each

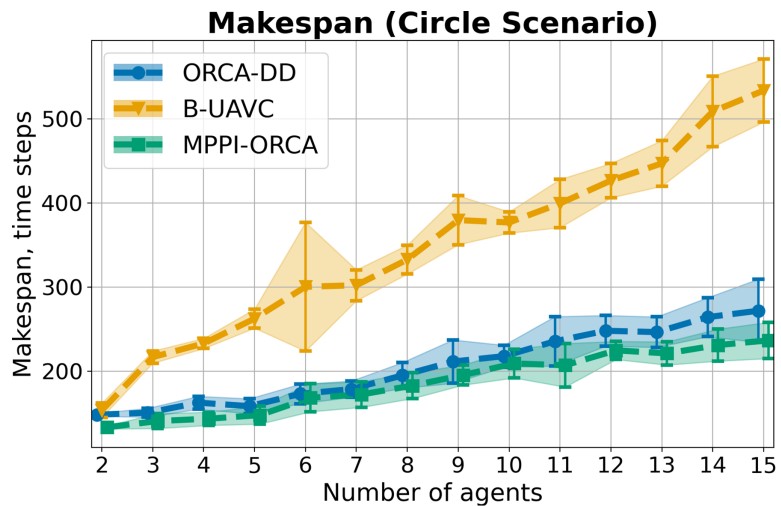

**Figure 6 The average *makespan* values and its standard deviation for evaluated algorithms for different number of agents in Circle scenario.** The lower is the better.

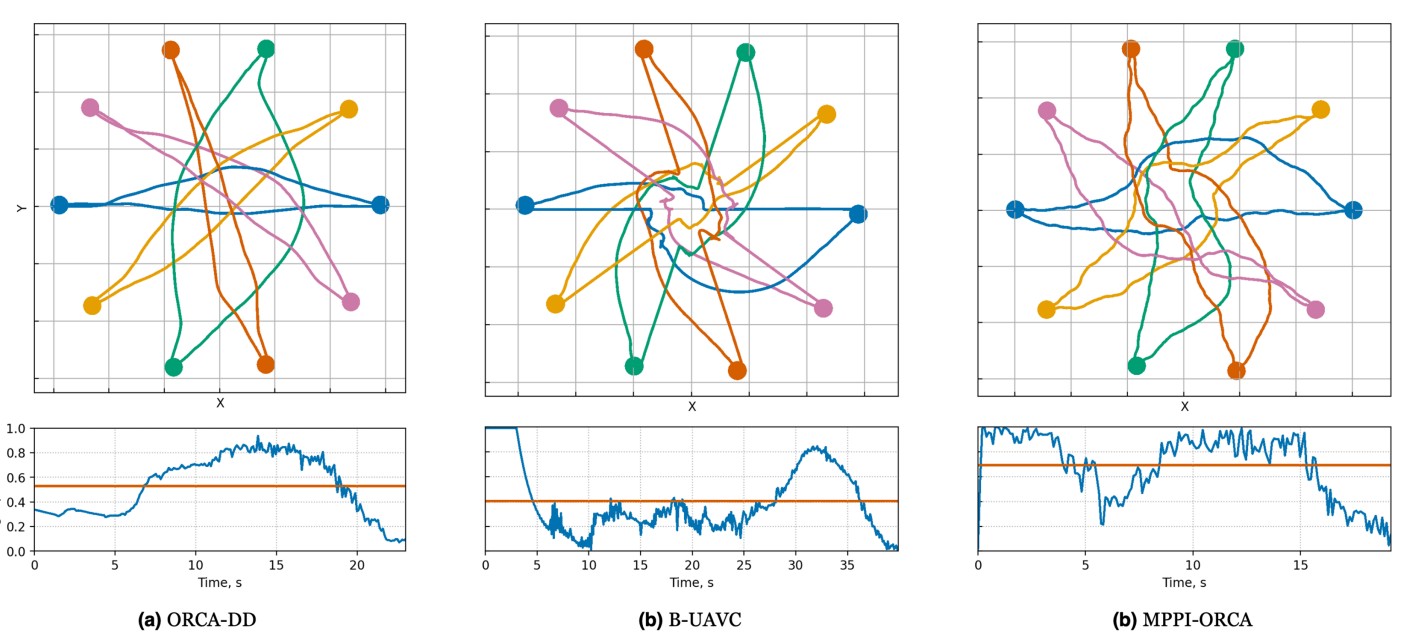

**(a)** ORCA-DD         **(b)** B-UAVC         **(b)** MPPI-ORCA

**Figure 7 Illustration of solutions for Circle instance with 10 agents for evaluated algorithms.** At the top, the trajectories visualization is shown. Below is a graph of the average agents' speed over time (blue line) and the average speed of all agents for the entire time of the simulation (brown line).

density, 10 instances were generated, each of which was launched 10 times (100 runs in total). Hence, the minimum decrement in the *success rate* is 1%. For example, the failure of one launch of one instance leads to a *success rate* of 99%. It can be observed that for tasks with four agents, all algorithms successfully find a solution in most cases, even with increasing density. However, when the number of agents increases to nine, the density

**Table 1 The average *success rate* values for evaluated algorithms for different density and different number of agents in `Grid` scenario.** Bold indicates the highest success rate values for a specified density and number of agents.

| Agents num. | Task type | ORCA-DD | B-UAVC | MPPI-ORCA |
|---|---|---|---|---|
| 4 | Sparse | 98% | **100%** | **100%** |
| | Medium | 97% | 99% | **100%** |
| | Dense | 87% | 90% | **100%** |
| 9 | Sparse | 93% | **100%** | **100%** |
| | Medium | 64% | 83% | **100%** |
| | Dense | 30% | 57% | **99%** |
| 16 | Sparse | 78% | 96% | **100%** |
| | Medium | 29% | 35% | **100%** |
| | Dense | 0% | 16% | **93%** |

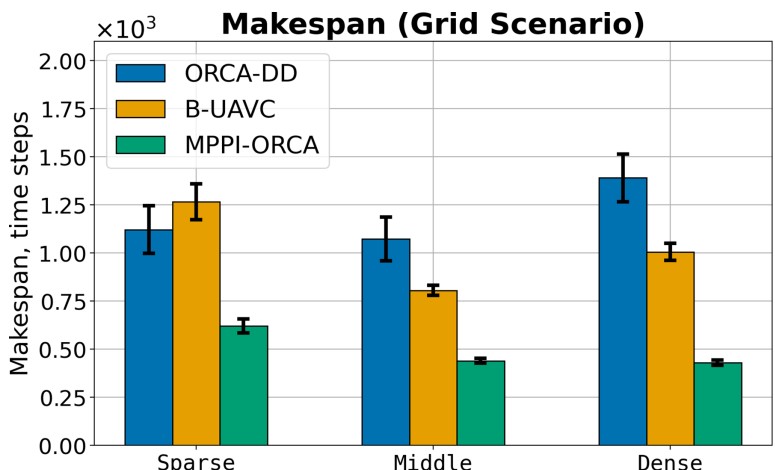

**Figure 8 The average *makespan* values and its standard deviation for evaluated algorithms for different density of agents in `Grid` scenario.** The result on a specific instance was added to the sum only if all the algorithms successfully solved it in all launches. The lower is the better.

significantly impacts the results of the ORCA-DD and B-UAVC algorithms. For example, with a minimum density of nine agents, ORCA-DD found a solution in 93% of launches, and the B-UAVC algorithm found a solution in 100% of launches. Yet, with a maximum density, the number of successful launches drops to 30% and 57%, respectively. Meanwhile, the proposed method copes successfully with a noticeably larger number of instances, and even for the most complex case with 16 agents and maximum density, the *success rate* drops only to 93%.

Let us consider the costs of obtained solutions in the `Grid` scenario, grouped by the density of agents in tasks, which is shown in Fig. 8. The plot illustrates the average *makespan* value among instances with the same density and standard deviation. If at least one instance launch of at least one algorithm failed, the instance was excluded from averaging. It can be seen that in all cases, the proposed method obtains more effective

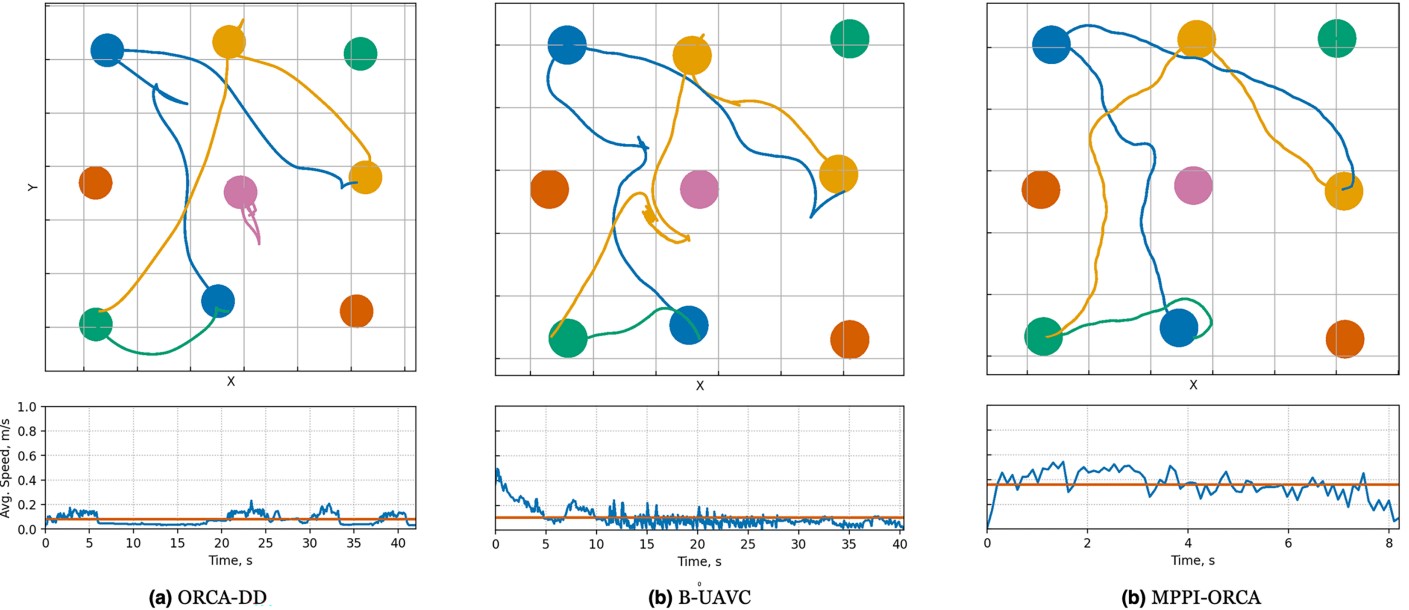

**Figure 9** **Illustration of solutions for `Grid-Sparse` instance with nine agents for evaluated algorithms.** At the top, the trajectories visualizationis shown. Below is a graph of the average agents' speed over time (blue line) and the average speed of all agents for the entire time of the simulation (brown line).

**Table 2** **The average *success rate* values for evaluated algorithms for different number of agents in `Random` scenario.** Bold indicates the highest success rate values for a specified number of agents.

| Agents num. | ORCA-DD | B-UAVC | MPPI-ORCA |
|---|---|---|---|
| 5 | **100%** | **100%** | **100%** |
| 10 | **100%** | **100%** | **100%** |
| 15 | 99.2% | 99.6% | **100%** |
| 20 | 99% | 99% | **100%** |
| 25 | **99.8%** | 97.8% | **99.8%** |

solutions with a noticeably smaller deviation. For example, for `dense` instances, the solution cost of the *MPPI*-ORCA algorithm is, on average, 3.2 times lower than the ORCA-DD and 2.3 times lower than the B-UAVC values. At the same time, unlike the `Circle` scenario, the duration of solutions of ORCA-DD is generally worse or comparable to the duration of solutions of B-UAVC. This can be explained by the fact that with a dense placement of agents, an increase in the radius of the agent begins to have a significant impact. An illustration of solutions of the `Sparse` instance with nine agents is given in Fig. 9.

Lastly, it is noteworthy that the average *makespan* for *MPPI*-ORCA decreases with increasing density. This is due to the fact that when averaging denser instances, fewer instances with a large number of agents (nine or 16) are included in the sample.

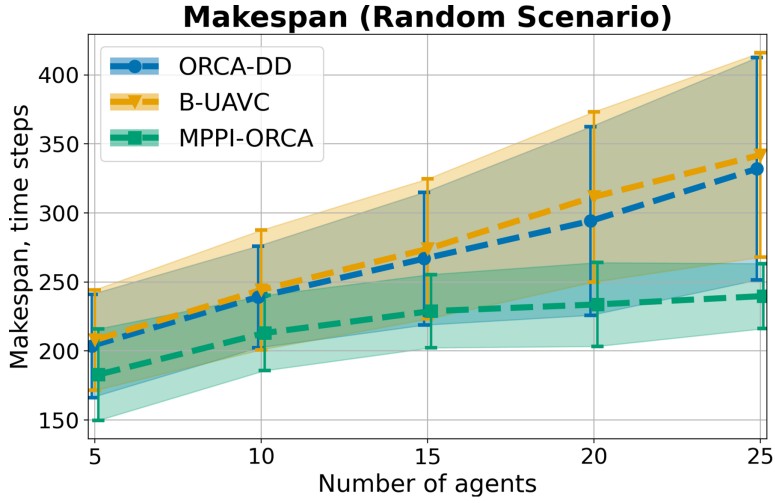

**Figure 10 The average *makespan* values and its standard deviation for evaluated algorithms for different number of agents in `Random` scenario.** The result on a specific instance was added to the sum only if all the algorithms successfully solved it in all launches. The lower is the better.

The average *success rate* of `Random` scenario launches, grouped by the number of agents, is shown in Table 2. According to the results from the table, all algorithms successfully coped in most cases, however, individual failed launches began to appear for ORCA-DD and B-UAVC algorithms with 15 agents, while the proposed method failed only once on an instance with 25 agents.

Let us consider the *makespan* values in `Random` scenario. The Fig. 10 display the average *makespan* value among instances with the same density and standard deviation. Similar to the `Grid` scenario, if at least one instance launch of at least one algorithm failed, the instance was excluded from averaging. The results indicate that the proposed algorithm, as in the previous scenarios, creates solutions of shorter duration on average. Moreover, the deviation of values is much lower than that of the other two methods. At the same time, ORCA-DD and B-UAVC show similar results in this scenario. Illustration of solutions of instance with 20 agents is given on Fig. 11.

**Average distance.** In addition to the metrics of *success rate* and *makespan*, the average distance traveled by agents was also measured. The results corresponding to the `Circle`, `Grid`, and `Random` scenarios are presented in Figs. 12–14.

Consider the results for the `Circle` scenario (Fig. 12). It is evident that, for all instances, the ORCA-DD algorithm generates shorter solutions in terms of distance compared to our method. Conversely, our method demonstrates superior performance compared to the B-UAVC algorithm.

However, the previously discussed solution durations indicate that our method achieves the goals faster on average than both competing algorithms. This may be attributed to the fact that the ORCA-DD method significantly reduces the speed of the agents when navigating complex situations, allowing some agents to overtake others. In contrast, the

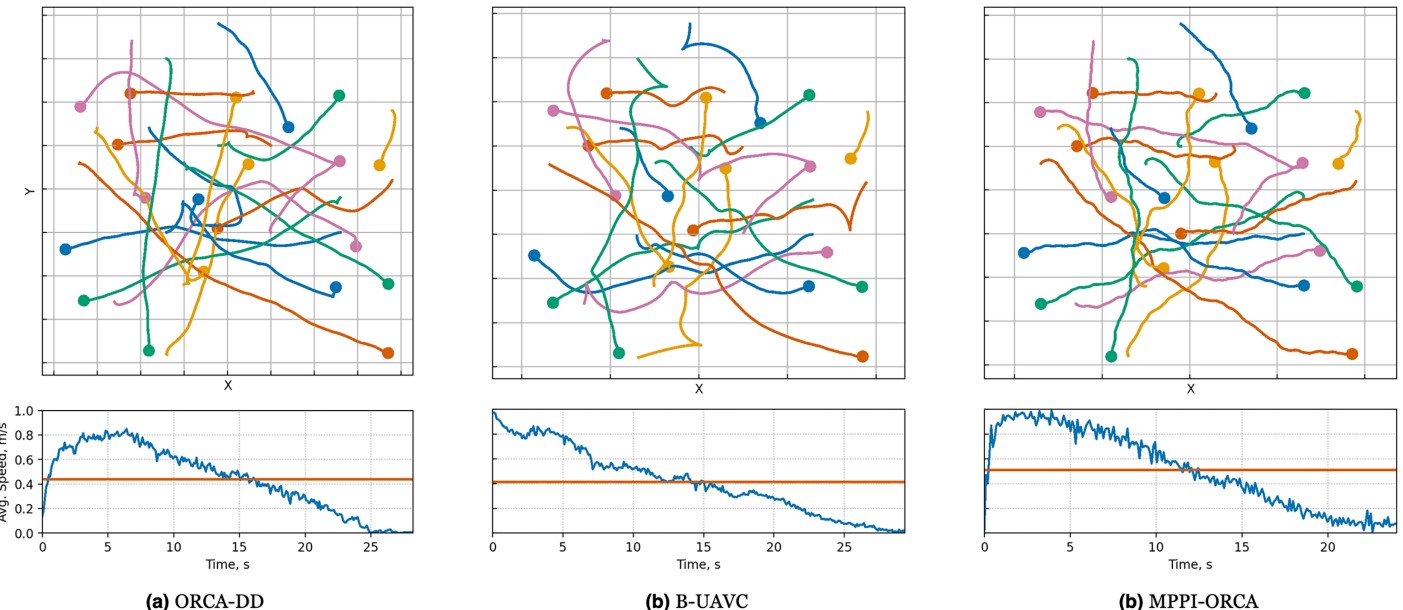

**Figure 11 Illustration of solutions for `Random` instance with 20 agents for evaluated algorithms.** At the top, the trajectories visualization is shown. Below is a graph of the average agents' speed over time (blue line) and the average speed of all agents for the entire time of the simulation (brown line).

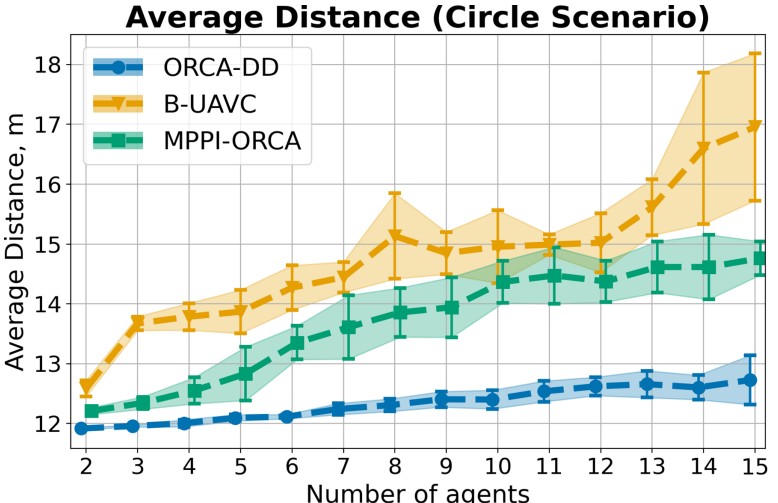

**Figure 12 The average travelled distance and its standard deviation for evaluated algorithms for different number of agents in `Circle` scenario.** The result on a specific instance was added to the sum only if all the algorithms successfully solved it in all launches. The lower is the better.

B-UAVC algorithm generates significantly oscillating trajectories when the agents are densely packed, resulting in prolonged solution durations and greater distances traveled.

This phenomenon is illustrated in Fig. 7, where the average speed of ORCA-DD agents decreases significantly during the time interval from 0 to 5, while the trajectories of most

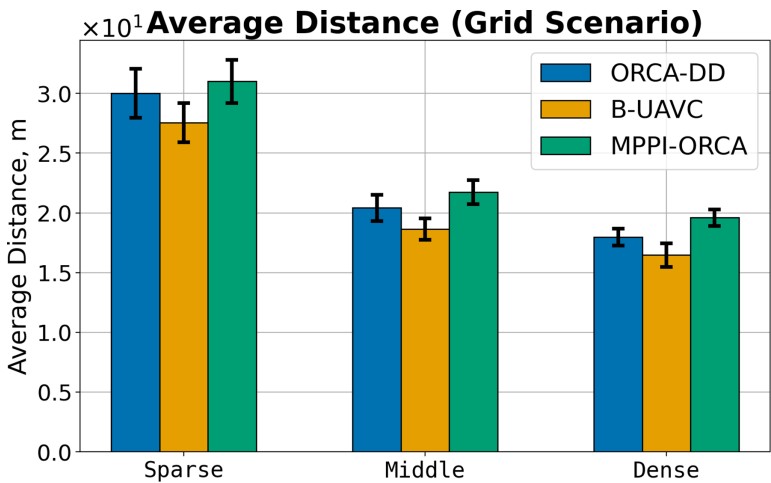

**Figure 13 The average travelled distance and its standard deviation for evaluated algorithms for different density of agents in `Grid` scenario.** The result on a specific instance was added to the sum only if all the algorithms successfully solved it in all launches. The lower is the better.

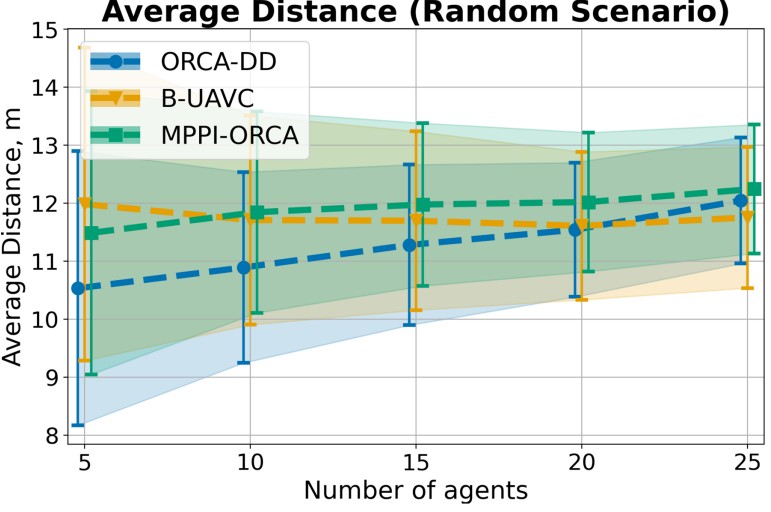

**Figure 14 The average travelled distance and its standard deviation for evaluated algorithms for different number of agents in `Random` scenario.** The result on a specific instance was added to the sum only if all the algorithms successfully solved it in all launches. The lower is the better.

ORCA-DD agents remain relatively straight. Additionally, the B-UAVC agents exhibit low mean velocity during the time interval from 5 to 25, with significant oscillation around the center of the circle.

In the `Grid` scenario, both competing algorithms produce solutions that are nearly equal to or slightly shorter than those generated by the proposed method. The solutions of B-UAVC are generally shorter than those of ORCA-DD. This may be attributed to the inflated radius of the ORCA-DD agent, which results in longer trajectories in dense environments.

We hypothesize that, similar to the `Circle` scenario, the proposed method is more proactive in resolving dense situations. In contrast, the other two methods tend to reduce agent speeds, resulting not only in longer solution durations but also in deadlocks. This behavior consequently leads to a lower percentage of solved problems. This hypothesis is further confirmed by the plots and illustrations in Fig. 9.

Last but not least, in the `Random` scenario, all three algorithms exhibit very similar results in terms of trajectory lengths, both in terms of mean values and overlap in standard deviations. However, the solution durations of the proposed algorithm are shorter than those of the competing algorithms (Fig. 10).

This may be attributed to the fact that, in this scenario, the agents are not densely located, leading to fewer conflict situations and trajectories that are nearly straight for all algorithms. However, the proposed method tends to select higher speeds on average, as evidenced by the speed plots in Fig. 11.

In general, the experimental results show that the proposed approach not only successfully copes with classical multi-agent navigation scenarios, but is also able to generate effective solutions in scenarios where the density of agents is high and the probability of a deadlock is high. At the same time, the resulting solutions have, on average, a lower duration than the two well-known collision avoidance algorithms, both in complex scenarios and in scenarios with a low density of agents. The distance travelled is on average close to similar indicators of competing methods, and increases relative to them only in dense scenarios.

Thus, based on the results for both duration-based and distance-based metrics, we suggest that the proposed method solves complex situations more efficiently by more actively overcoming deadlocks, which leads to longer trajectories, but allows agents to reach goals faster on average.

## Car-like dynamics

### *Experimental setup*

The second series of experiments consisted in validating the proposed method when controlling car-like robot. For these purposes, the next model was used. The robots' state $\mathbf{x} = (p_x, \ p_y, \ \theta)^T$ in this case is the same as for a differential-drive robot. However, the control input $\mathbf{u} = (v, \ \phi)^T$ in this case is the linear velocity $v$ and the steering angle $\phi$. At the same time, similarly to the differential-drive case, the range of control actions is bounded ($v_{min} \le v \le v_{max}$, $-\pi/2 < \phi_{min} \le \phi \le \phi_{max} < \pi/2$). The equations of motion involved in model was the following:

$$p_{x,t+1} = p_{x,t} + v_t \cos \theta_t \Delta t \tag{33}$$
$$p_{x,t+1} = p_{y,t} + v_t \sin \theta_t \Delta t \tag{34}$$
$$\theta_{t+1} = \theta_t + \frac{v}{L} \tan \phi_t \Delta t \tag{35}$$

where $L$–distance between rear and front axles. It is easy to see, that in this case, agents can not turn around when linear velocity $v$ is zero. Thus, the dynamics of this type significantly complicates collision avoidance.

**Table 3 The *success rate* and average *makespan* values for suggested approach in Circle scenario for car-like dynamics.**

| Agents num. | 2 | 3 | 4 | 5 | 6 | 7 | 8 | 9 | 10 | 11 | 12 | 13 | 14 | 15 |
|---|---|---|---|---|---|---|---|---|---|---|---|---|---|---|
| Success rate | 100% | 100% | 100% | 100% | 100% | 100% | 100% | 100% | 100% | 100% | 100% | 100% | 100% | 100% |
| Makespan | 166.8 | 169.8 | 193.4 | 211.2 | 216.8 | 247.0 | 276.6 | 285.4 | 298.7 | 319.4 | 342.0 | 355.5 | 368.7 | 392.7 |

The following parameters for all robots were used in the experiments: robot sizes (radius of the corresponding disks) $r = 0.3\ m$, linear velocity limits $v_{min} = -1.0\ m/s$, $v_{max} = 1.0\ m/s$, angular velocity limits $\phi_{min} = -\frac{\pi}{3}\ rad/$, $v_{max} = \frac{\pi}{3}\ rad$, the distance between axles $L = 0.2$, sight/communication radius $R$ was not limited.

In the experiment, the Circle scenario was involved with the same instances as in experiments with differential-drive robots. The values of $\Delta t$, $\varepsilon$ and $k_{lim}$ were set to 0.1 s, 0.3 m and 1,000 *steps*, respectively. The additional safety buffer $\varepsilon_r = 0.05\ m$ as in differential-drive case was also used. Each instance was launched 10 times.

### Experimental results

The main metrics used during the experiment were also *success rate* and *makespan*. The results of launches of the suggested approach are shown in Table 3. It can be seen, that in all cases our approach successfully solve all task at all launches.

At the same time, we note that the *makespan* has grown relative to the differential-drive dynamics. For example, for two agents in the scenario, the value has increased by about 25%, and for 15 by 66%. This growth may be due to the fact that, unlike differential-drive robots, car-like dynamics does not imply the possibility of turning in place. Thus, with a dense arrangement of agents, they need to either move back, or, in some cases, even move along the loop.

Such effects can be observed on the visualization of trajectories on Fig. 15 obtained during the execution of tasks. So, it can be observed that for a few agents (2 and 5), the trajectories are smooth and similar to trajectories of differential-drive robots, but for cases with 10 and 15 agents, the trajectories include "jiggling" and loops.

### Comparison with learning-based approach
#### Experimental setup

In this experiment, the proposed approach was compared with modern reinforcement learning-based method (*Blumenkamp et al., 2022*). The original implementation provided by the authors of the article was used for training and launches. However, for the purposes of the experiment, a number of changes were made to the setup.

The main change was the exclusion of obstacles from the environment. We also eliminated the acceleration restrictions for the learning-based method. In addition, the following train and experimental parameters were used. The suggested approach used differential-drive motion model (Eq. (3)). Robot sizes (radius of the corresponding disks) $r = 0.25\ m$, linear velocity limits $v_{min} = -1.0\ m/s$, $v_{max} = 1.0\ m/s$, angular velocity limits $w_{min} = -2.0\ rad/s$, $w_{max} = 2.0\ rad/s$, sight/communication radius $R$ was not limited. We also involved additional safety buffer of size $\varepsilon_r = 0.05\ m$ for both algorithms

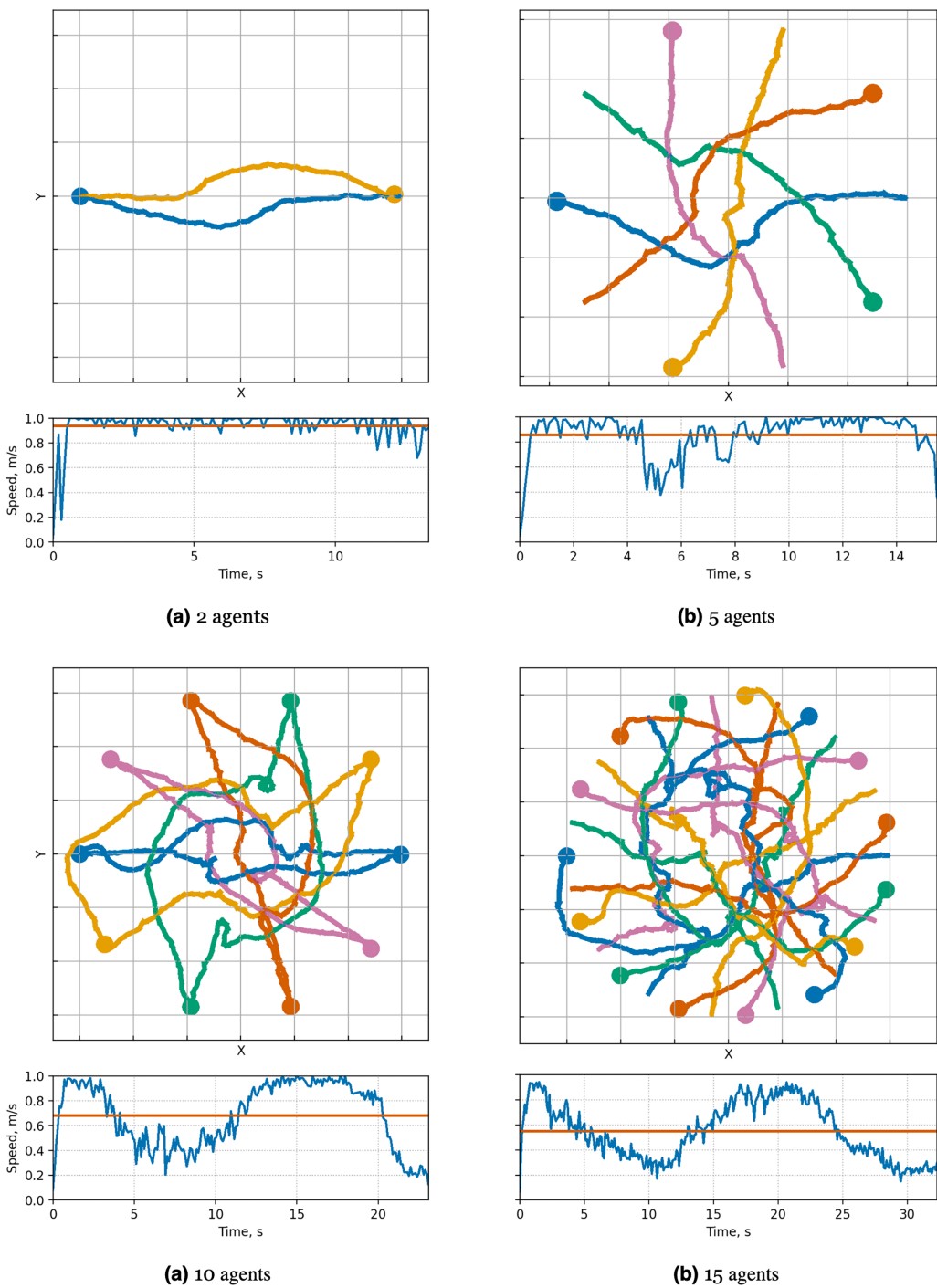

**Figure 15** **Illustration of solutions for `Circle` instance with 2, 5, 10, 15 car-like agents for MPPI-ORCA algorithms.** At the top, the trajectories visualization is shown. Below is a graph of the average agents' speed over time (blue line) and the average speed of all agents for the entire time of the simulation (brown line).

to minimize the chance of collision. The values of $\Delta t$, $\varepsilon$ and $k_{lim}$ were set to 0.1 $s$, 0.3 $m$ and 1,000 $steps$, respectively. Each instance in experiment was launched 10 times. The scenario proposed by the authors of the work, including five agents, was used for training.

**Table 4 The average *success rate* values for suggested algorithm in comparison with learning-based approach for different number of agents.** Bold indicates the highest success rate values for a specified number of agents.

| Agents num. | MPPI-ORCA | Multi-Agent RL |
|---|---|---|
| 4 | **98%** | 48% |
| 5 | **98%** | 42% |
| 6 | **87%** | 16% |

It is worth noting separately that, unlike the proposed approach, the trained method uses a holonomic model of motion (*i.e.*, such a model of motion where the agent chooses the desired velocity at each step, after which it perfectly executes it; at the same time, the agent is able to instantly change both the amplitude of the velocity and its direction). Thus, the limit on the maximum angular velocity was ignored. This is dictated by the fact that when training agents with differential-drive dynamics, the method did not allow finding a solution for training scenarios even after 4,000 training iterations.

In this scenario, the initial and goal positions of the robots were located similarly to the `Circle` scenario described earlier. The radius of the circle was set to 0.7 *m*. However, when generating instances, the center positions of the circles for the initial and goal positions were placed at a random distance from each other. In addition, all positions in the initial and goal circles were shifted to a random angle (separate angles for the initial and goal positions were used). The number of agents ranged from four to six with step 1, and 50 different instances were generated for each number of agents.

### Experimental results

The main metrics measured during the experiment were the *success rate*. The average *success rate* of launches of instances, grouped by the number of agents, is shown in Table 4. In the table, it can be seen that even when using a more complex dynamic model, the proposed approach shows noticeably better results in all cases. Separately, we note that the scenario chosen for this series of experiments was designed in such a way as to be similar to the scenario used in training. When using other scenarios, the success rate of the learning-based method decreased significantly. We assume that this is not due to the disadvantages of the method, but to the limitations of the training scenarios. However, this is an indicator of the difficulties associated with the generalization abilities of the learning-based approaches.

Illustration of solutions for suggested and learning-based methods in instances with four, five, and six agents for evaluated algorithms are shown in Fig. 16. It can be seen that in scenarios where the learning-based algorithm found solutions, they can be more straightforward, however, this is due to the fact that the proposed algorithm takes into account the differential-drive constraints while the learning-based algorithm does not.

## Heterogeneous agents and information uncertainty
### Experimental setup

In this scenario, we decided to examine the possibilities of the proposed method to operate under conditions of uncertainty in the available information and to estimate the influence

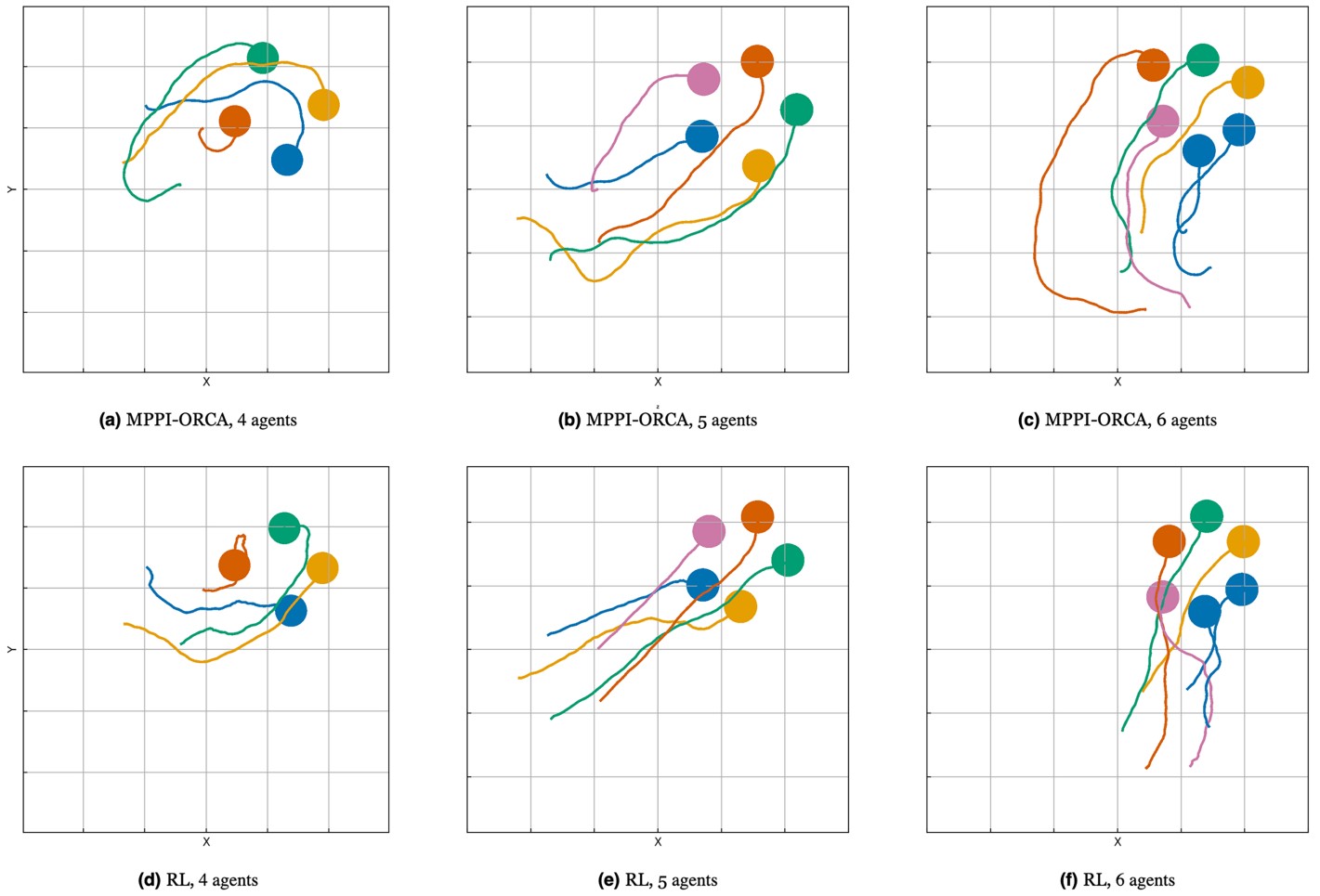

**Figure 16** **The trajectories visualization for suggested (A–C) and learning-based (D–F) methods with 4, 5, 6 agents for evaluated algorithms.**

of uncertainty on the results. In addition, it was decided to check the ability to operate with different agents moving in the same environment.

For these purposes, the modified `Random` scenario with differential-drive robots was used. All the agents in the scenario were divided into two groups. In the first group, all the robots had a radius of 0.2 $m$, linear velocity limits $v_{min} = -1.0$ $m/s$, $v_{max} = 1.0$ $m/s$, angular velocity limits $w_{min} = -2.0$ $rad/s$, $w_{max} = 2.0$ $rad/s$. In the second group, all robots had a radius of 0.5 $m$, linear velocity limits $v_{min} = -3.0$ $m/s$, $v_{max} = 3.0$ $m/s$, angular velocity limits $w_{min} = -6.0$ $rad/s$, $w_{max} = 6.0$ $rad/s$. In addition, the safety buffer has been increased to 0.15 $m$ to deal with inaccuracies in the input data.

Uncertainty was added to the data as follows. At each step of the algorithm, modified information about its position, velocity and direction, as well as the positions and velocities of other agents, was provided as input data. Each value was modified by normal noise with specified standard deviations ($\sigma_{xy}$ for position noise, $\sigma_v$ for velocity noise, $\sigma_\theta$ for direction noise). The following values of uncertainty were used in the experiments. The first step was

**Table 5 The average *success rate* values for suggested algorithm with varying levels of uncertainty in the available data for different number of agents in `Random` scenario with heterogeneous agents.** Bold indicates the highest success rate values for a specified number of agents.

| N | Perfect | $\sigma_{xy} = 0.01, \sigma_\theta = 0.02$ | $\sigma_{xy} = 0.08, \sigma_\theta = 0.04$ | $\sigma_{xy} = 0.16, \sigma_\theta = 0.08$ |
|---|---|---|---|---|
| 5 | **100%** | **100%** | **100%** | **100%** |
| 10 | **100%** | **100%** | **100%** | 99% |
| 15 | 98% | 99% | **100%** | 84% |
| 20 | 94% | 97% | **100%** | 47% |
| 25 | 93% | 96% | **99%** | 12% |

to run an algorithm without inaccuracy in the information. Next, three levels of uncertainty were used. In the first level, the values were set to $\sigma_{xy} = 0.01, \sigma_\theta = 0.02$, in the second level $\sigma_{xy} = 0.08, \sigma_\theta = 0.04$, in the third level $\sigma_{xy} = 0.16, \sigma_\theta = 0.08$. The level of uncertainty in the velocity data was set based on the level of uncertainty in the position

$$\sigma_v = \sqrt{\sigma_{xy}^2 + \sigma_{xy}^2}.$$

### Experimental results

The main metrics measured during the experiment were *success rate* and *makespan*. The average *success rate* of launches, grouped by the number of agents, is shown in Table 5. In all launches except experiments with the highest level of uncertainty, the drop in success rate is associated only with exceeding the time step limit. However, in experiments with the highest level of uncertainty, isolated cases of agent collisions were observed (one case for tasks with 15 and 20 agents, and four cases for tasks with 25 agents). Moreover, increased uncertainty also hinders agents from accurately reaching the goal area, which can also influence the results, especially with a high level of inaccuracy in the data.

According to the results from the table, we can observe that for all levels of uncertainty, except the highest, the algorithm copes with most of the tasks. Consequently, it can be concluded that the proposed approach is able to successfully handle heterogeneous scenarios in which a limited level of uncertainty is allowed. However, if the uncertainty is significantly high, it is necessary to select a suitable safety buffer to prevent solutions that may lead to collisions. Nevertheless, an excessive increase in the radius of the agent can lead to deadlocks. It is also important to consider that inaccuracy in the information may result in the inability to accurately reach the goal position.

Furthermore, it is interesting to note an effect where, with a slight increase in uncertainty, the proportion of successful solutions may increase. A hypothesis is that a small amount of randomness in the data makes it easier to overcome deadlocks or symmetric situations, similar to how the goal vector for ORCA-DD and B-UAVC methods was modified by a small random value.

The average *makespan* and standard deviation for each number of agents are shown in Fig. 17. The results took into account instances that were successfully solved at least once at all levels of uncertainty. Unsuccessful launches were removed from the sample and did not affect the average. From the results, it can be seen that uncertainty can affect the quality of

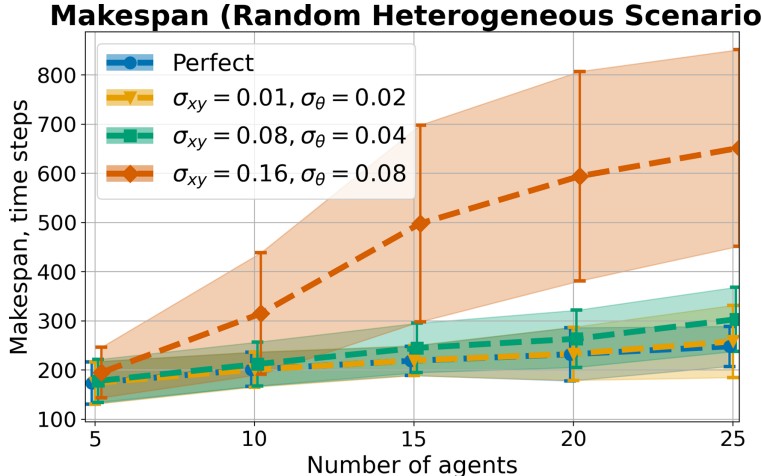

**Figure 17 The average *makespan* values and its standard deviation for suggested algorithm with varying levels of uncertainty in the available data for different number of agents in** `Random` **scenario with heterogeneous agents.** The lower is the better.

the solutions obtained, and with the introduction of a small inaccuracy in the data, the quality of solutions also changes slightly. However, the quality of solutions and their stability are significantly reduced if the uncertainty is high.

The experiments have shown that the proposed method is capable of performing complex tasks involving agents of various types, even when there is uncertainty in knowledge about the state of the agent and its neighbors. At the same time, when choosing algorithm parameters, it is worth taking into account the level of uncertainty in the data that may occur under certain conditions.

## CONCLUSIONS

This article addresses the issue of decentralized multi-agent collision avoidance and proposes a sampling-based method to solve this problem that incorporates kinematic constraints. The key concept of the proposed approach involves determining a distribution based on linear constraints, enabling the sampling of safe control actions. We demonstrate that, in general, the distribution parameters can be obtained by solving a *Second-Order Cone Programming* (*SOCP*) optimization problem. We establish that the obtained solutions are not only safe, but also enhance sampling efficiency.

Subsequently, we conduct a comprehensive evaluation of the proposed method through a series of experiments encompassing diverse scenarios involving the differential-drive robot and car-like dynamics. A comparative analysis is performed against state-of-the-art decentralized collision avoidance methods, namely *differential-drive ORCA* and *B-UAVC*. The results illustrate the efficacy of the suggested approach as it successfully solves challenging tasks, including scenarios in which existing algorithms fail to find appropriate solutions. Furthermore, our method demonstrates improved solution quality, particularly in terms of duration, compared to other methods.

We also compared the proposed approach with a modern learning-based method and showed that, on the one hand, the proposed method copes with the task more effectively. On the other hand, the results show that the learning-based methods can experience significant difficulties with generalization, especially if the learning scenarios cover only a small number of situations.

Finally, we conducted experiments with heterogeneous agents involving uncertainty in the available information. The results showed that, with the right choice of parameters, our method effectively copes with cases of limited uncertainty in the data.

Future work will explore in more detail the incorporation of uncertainties in state and neighbor information, as well as inaccuracies in control execution. Additionally, an important direction for further research involves the implementation of the proposed approach using the Robot Operating System (ROS) (*Macenski et al., 2022*), and conducting experiments utilizing more advanced simulations such as Gazebo (*Koenig & Howard, 2004*) or CoppeliaSim (*Rohmer, Singh & Freese, 2013*), and also on real robots.

### Funding
This work was supported by the Ministry of Science and Higher Education of the Russian Federation (Project No. 075-15-2024-544). The funders had no role in study design, data collection and analysis, decision to publish, or preparation of the manuscript.

### Grant Disclosures
The following grant information was disclosed by the authors:
Ministry of Science and Higher Education of the Russian Federation: 075-15-2024-544.

### Competing Interests
The authors declare that they have no competing interests.

### Author Contributions
- Stepan Dergachev conceived and designed the experiments, performed the experiments, analyzed the data, performed the computation work, prepared figures and/or tables, authored or reviewed drafts of the article, and approved the final draft.
- Konstantin Yakovlev conceived and designed the experiments, analyzed the data, authored or reviewed drafts of the article, and approved the final draft.

### Data Availability
The source code of experimental implementation for suggested method is available at GitHub and Zenodo:
- https://github.com/PathPlanning/MPPI-Collision-Avoidance.
- Stepan Dergachev. (2024). PathPlanning/*MPPI*-Collision-Avoidance: PeerJ CS Resubmit (May 2024) (v0.2.1). Zenodo. https://doi.org/10.5281/zenodo.11112262.

## Supplemental Information

Supplemental information for this article can be found online at http://dx.doi.org/10.7717/peerj-cs.2220#supplemental-information.

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
