# Peer review of "Model predictive path integral for decentralized multi-agent collision avoidance"

_PeerJ Computer Science, doi:10.7717/peerj-cs.2220_

## Round 0.1 · original submission · Major Revisions

The reviewers agree on the importance of the topic and the innovativeness of the proposed approach. However, the experimental activity presented is not complete enough to support the conclusions. In particular, more realistic experiments and a comparison with other approaches should be added. Please refer to the reviews for the details.

**Language Note:** The review process has identified that the English language must be improved. PeerJ can provide language editing services - please contact us at copyediting@peerj.com for pricing (be sure to provide your manuscript number and title). Alternatively, you should make your own arrangements to improve the language quality and provide details in your response letter. – PeerJ Staff

Reviewer 1 ·

Basic reporting

The paper presents a decentralized approach to collision avoidance in a multi-robot context.

The proposed approach is based on existing frameworks, like Model Predictive Path Integral (MPPI) and Optimal Reciprocal Collision Avoidance (ORCA), and the original contribution of the paper mainly consists of adapting these frameworks to the setting considered in the paper.
One of the advantages of the proposed approach is that it is agnostic with respect to the locomotion model of the robots.


Generally, the paper is well-written and clear in illustrating the pipeline of the proposed approach.
However, some aspects of the technical presentation should be improved:
- What is the bar in the formula just after (4)?
- How does the parameter \gamma in (14) relate to the parameter \lambda in (8)? More generally, it is not completely clear what is the relationship between (8) and (13)-(14).
- Page 17/29: 5etter -> better
- It is unclear over how many runs are the percentages of Table 1 (and of other tables) calculated. I understood that 10 runs were performed for each instance (= row of the table?) but then I cannot figure out why the success rate can be 99%.

Experimental design

Experiments:
- It is assumed that the setting is noise-free. While this is ok for the theoretical derivation, it would be interesting to see how the proposed method performs in practice when noise affects sensing and control actions. Please add experiments to address this issue.
- It would be also interesting to see what happens when robots are heterogeneous both in size and in velocity bounds. Please add experiments to address this issue.
- All the experiments are conducted in environments without obstacles. However, obstacles are present in several 2D environments in which multi-robot systems are employed (like in warehouses). How does the proposed method (and the alternative methods it is compared against) perform in the presence of obstacles? Please add experiments.
- For car-like dynamics, why there are no comparisons with other methods? Why are results only reported for only one (Circle) scenario?

Validity of the findings

A stronger experimental evaluation can better support the argument of the validity of the proposed approach.

Cite this review as

·

Basic reporting

The authors propose a new collision avoidance algorithm in a multi-agent decentralized setting. They enhance and adapt the Model Predictive Path Integral based trajectory estimation by incorporating the Optimal Reciprocal Collision Avoidance based linear constraints estimated from multi-agents's state information. They propose to solve the resultant non linear optimization problem by reducing it to Second Order Code program.

The paper is clearly written. However, there are some minor grammatical errors that crept in. Below are the sentences that need to be correct. Please search them in the paper and correct them.

Line 150 and 151: When writing about existing works, please just dont use "Works" but instead use a better way to refer them, like in line 164 or other parts of the paper.

Line 183: B-UAVC constraints. Which requires the -> This sentence either needs to be combined or re-written

Line 356: communicate, chosen action not guarantee, that -> I think it should be "chosen action cannot guarantee"

Line 363: "reduce" should be "reduction" I think

Line 414: Should it be Program or Programm ?

Line 550: It should be "better" and not "5etter"

At this stage, the paper proposes a method that has assumptions which do not hold true in real world.

It is recommended to present some overview or literature survey on active collision avoidance systems, which is heavily used in real world multi-robot or single robot deployment. Also the ROS or ROS2 Nav Stack already has an active collision avoidance system and a comparision with it is expected, and highlight how the proposed method is different.

This is very unrealistic assumption, however for research purposes, I am okay with it. However, I expect to see how the system shall behave when this is not met and there is some noise.
Line 237: "each action is executed perfectly, so at each time step, an agent’s position is deterministic and known exactly"

Experimental design

The experiments are only simulation based and does not consider noise in sensor readings or tranmission or delay in computation in multi-agent setup. Theoretically, it is acceptable but it is highly encouraged to vary some of these parameters and report some initial results.

The proposed method is compared with only two base line algorithms based on which the proposed method is improved. It is expected that the proposed method outperforms them. I highly encourage authors to add some experiments with other methods apart from ORCA-DD and B-UAVC, so that the proposed method can be compared with the existing state of the art.

What is the existing state of the art or the toughest competitor for the proposed algorithm ? Is it possible to compare atleast with one or two of such algorithms ?

If such comparison is not possible, I encourage authors to reason why.

I would like to see what is the improvement that the system gains in these two conditions:
1. Without converting the non-linear optimization to SOCP and directly solving that non-linear equation
2. Solving the problem as SOCP.

What are the advantages of step 2 over step 1 ?

Some estimate of the computational requirements and efficiency reporting is encouraged.

Validity of the findings

At the end, these systems are aimed for real world deployment. Though this journal does not judge based on impact or novelty, it really cares about reproducibility.

It is highly recommended that the authors should open-source their C++ implementation with a toy example and relevant sample data to run basic tests of their algorithm. There can be many parameters that were not discussed in the paper and may have significant effect on the performance of the system.

It is highly recommended that authors should consider noise in sensor readings, estimation and transmission and provide some simulation results in this setup. How does noise in each robot's motion or sensing of other robot’s velocity change the accuracy of the proposed approach ? Does the system still work or will it fall apart ? In any case, just reporting the simulation results is highly encouraged.

---

## Round 0.2 · Minor Revisions

The reviewer believes the manuscript has significantly improved compared to the previous version. However, there are still a few minor issues that need to be addressed before the paper is ready for publication. Please refer to the reviewer's comments for details.

Reviewer 1 ·

Basic reporting

The paper has definitely improved with respect to the previous version.

Experimental design

Beyond makespan (which roughly measures the efficiency of coordination), it would be also interesting to have results about the length of the trajectories followed by the robots, in order to have an idea about the cost of coordination. At least for the experiments on differential-drive dynamics.

Validity of the findings

Line 92: claiming that the MAPF algorithms “do not guarantee finding a solution” is misleading, especially after the statement that they often provide theoretical guarantees on the quality of the solutions. Authors should reformulate their claims.

The text of the paper refers to Appendices, but I could not find them. Please reformulate.

From a computational point of view, it sounds weird that the authors have an NP-hard problem (which is (25)) and translate it into a polynomially-solvable problem (which is (26)). Is the original problem (25) not so computationally hard, is the translation approach flawed, or are the polynomial algorithms used to find a solution for (26) not finding the optimal solution but just a solution? Authors should clarify these issues.

The neighborhood of the goal position is denoted differently in lines 232 and 513. Please make the notation consistent.

Typos:
- Just before Equation (20): satisfy -> satisfies
- Just before Equation (33): involve -> involved

Cite this review as

---

## Round 0.3 · accepted · Accept

In the last review round, the reviewer expressed only minor concerns, which mainly concerned the presentation. Since the reviewer only asked for minor revisions, I did not invite him back and I personally checked the revised version. I confirm that the authors have properly addressed the concerns and therefore believe that the paper is ready for publication.